# The Arabidopsis endosperm is a temperature-sensing tissue that implements seed thermoinhibition through phyB

Urszula Piskurewicz[1], Maria Sentandreu [1], Mayumi Iwasaki [1], Gaëtan Glauser[2] & Luis Lopez-Molina [1,3] ✉

Seed thermoinhibition, the repression of germination under high temperatures, prevents seedling establishment under potentially fatal conditions. Thermoinhibition is relevant for phenology and agriculture, particularly in a warming globe. The temperature sensing mechanisms and signaling pathways sustaining thermoinhibition are unknown. Here we show that thermoinhibition in *Arabidopsis thaliana* is not autonomously controlled by the embryo but is rather implemented by the endosperm. High temperature is sensed through endospermic phyB by accelerating its reversion from the active signaling Pfr form into the inactive Pr form, as previously described in seedlings. This leads to thermoinhibition mediated by PIFs, mainly PIF1, PIF3 and PIF5. Endospermic PIF3 represses the expression of the endospermic ABA catabolic gene *CYP707A1* and promotes endospermic ABA accumulation and release towards the embryo to block its growth. Furthermore, endospermic ABA represses embryonic PIF3 accumulation that would otherwise promote embryonic growth. Hence, under high temperatures PIF3 exerts opposite growth responses in the endosperm and embryo.

The onset of the embryo-to-seedling transition is a major event in the plant's life cycle as the plant abandons its embryonic resistant state within the mature seed to form a fragile seedling. Subsequent reproductive success will depend on whether the seedling survives the environment it confronts. It is therefore unsurprising that germination control mechanisms have evolved to improve the seedling's chances for survival.

Newly formed seeds exhibit primary seed dormancy (hereafter referred as dormancy), a trait whereby seeds do not germinate under otherwise favorable germination conditions[1]. Over time, seeds lose dormancy as they age (a process referred as dry after-ripening), i.e. they acquire the capacity to germinate under favorable conditions, thus becoming non-dormant. Dormancy delays the onset of the embryo-to-seedling transition to ensure that seedlings are formed during a favorable season. However, non-dormant seeds continue to control

their germination if confronted with non-favorable conditions. This includes the capacity to block seed germination under high temperatures, an adaptive trait referred as seed thermoinhibition. Seeds can be remarkably discerning when confronted with high temperatures and a difference of 1 °C may determine whether a substantial percentage of a seed population germinates or not[2]. Thermoinhibition furthers increases the odds of establishing a viable seedling since seedlings may not survive under persistent high temperatures. Hence, together with dormancy, thermoinhibition represents an additional control layer, enabling plants to form a seedling during a favorable season to conduct their reproductive phase. The trait is expected to have an impact on plant ecology, phenology and agriculture and this impact will be even greater as temperatures increase throughout the globe.

Thermoinhibition has been genetically studied in lettuce (*Lactuca sativa*) and *Arabidopsis thaliana*[3–5]. Thermoinhibition requires ABA

[1]Department of Plant Sciences, University of Geneva, Geneva, Switzerland. [2]Neuchâtel Platform of Analytical Chemistry, Université de Neuchâtel, Neuchâtel, Switzerland. [3]Institute of Genetics and Genomics in Geneva (iGE3), University of Geneva, Geneva, Switzerland. ✉e-mail: Luis.LopezMolina@unige.ch

synthesis, as mutants unable to synthesize ABA lack thermoinhibition, and is associated with low GA biosynthesis[2,3]. Toh et al. showed that *rgl2* mutants, lacking the GA response factor RGL2 have lower seed thermoinhibition although remaining responsive to high temperatures[2]. Previous work showed that low GA levels lead to stabilization of DELLA factors, such as RGL2, which promote ABA synthesis[6,7]. Thus, other DELLA factors accumulating in *rgl2* mutants could still promote thermoinhibition by promoting ABA synthesis. However, and interestingly, Toh et al also reported that exogenous GA can promote germination at very high temperatures (34 °C) but without fully abolishing thermoinhibition[2]. This suggests that there are additional parallel pathways promoting thermoinhibition independently of the GA signaling pathway. Furthermore, the sensing mechanisms enabling seeds to trigger thermoinhibition is unknown.

Like seeds, seedlings are capable of discerning small changes in temperature to adapt their growth rate and avoid heat stress. Increasing temperatures promote hypocotyl and petiole growth as well as flowering[8,9]. Phytochromes are a family of five photoreceptors in Arabidopsis (phyA, phyB, phyC, phyD and phyE) synthesized in a signaling-inactive state known as Pr. Upon absorption of red light (R), they convert into a signaling-active Pfr state. In its Pfr state phyB interacts with phytochrome interacting factors (PIFs), a family of basic helix-loop-helix transcription factors regulating photomorphogenic development, to promote their degradation. In absence of the Pfr state, PIFs are stabilized, which promotes their accumulation[10,11]. The Pfr state can revert to the Pr state upon absorption of far-red light (FR) or through thermal relaxation (also called "thermal reversion"), a process accelerated by increasing temperatures and occurring even in absence of light[10,11]. Phytochrome thermal reversion, and particularly that of phytochrome phyB, was proposed to be the underlying mechanism by which seedlings sense temperature[12,13]. In turn, a reduction of the Pfr pool promotes the stabilization of phytochrome-interacting factors (PIFs) that promote growth[10,11]. Previous work also showed that phyB inactivation by far red-light in the seed endosperm blocks germination[14]. phyB inactivation leads to increased endospermic ABA levels and release, which blocks germination[14,15]. This conclusion was notably reached using a "Seed Coat Bedding Assay" (SCBA) whereby dissected embryos are cultured on a bed of dissected endosperms, which enables to study embryonic growth under the influence of endosperm tissues[16]. The SCBA enables the use of embryo and endosperm of different genetic backgrounds to genetically dissect the processes regulating growth in the embryo and endosperm. However, whether phytochromes and the endosperm play a role in seed thermoinhibition is poorly understood.

Here, we show that thermoinhibition in *Arabidopsis thaliana* is dependent on the endosperm rather than the embryo as embryos deprived of their endosperm are unable to repress their growth under high temperatures. Using the SCBA and direct ABA measurements, we show that at high temperatures the endosperm releases ABA towards the embryo to block its growth and maintain its embryonic state by promoting the accumulation of the ABA response factor ABI5. In the endosperm, high temperature reduces the pool of active phyB, which increases endospermic ABA levels and release. This involves endospermic DELLA factors and PIFs, particularly PIF1, PIF3 and PIF5. We found that PIF3 represses the expression of *CYP707A1*, encoding an ABA catabolic gene[17,18]. During the embryo-to-seedling transition PIF3 strongly accumulates in the embryo and promotes early seedling growth. Under high temperatures endospermic ABA represses embryonic PIF3 accumulation, indicating that PIF3 exerts opposite growth responses in the endosperm and embryo under high temperatures.

## Results

### The endosperm enables seed thermoinhibition by releasing ABA
The endosperm promotes dormancy by releasing ABA towards the embryo upon imbibition. We explored whether the endosperm plays a role for seed thermoinhibition. Embryos deprived of their seed coat and endosperm 4 h after seed imbibition grew under high temperatures (34 °C), expanding their cotyledons and doubling their hypocotyl length 3 days after dissection (Fig. 1a, b). In contrast, embryos from intact seeds did not germinate and maintained their embryonic state (Fig. 1a, b). Furthermore, embryos remained thermoinhibited when only the seed coat was removed (Fig. 1c). These results show that presence of the endosperm is essential for seed thermoinhibition. Previous work showed that endogenous ABA synthesis is necessary for thermoinhibition but whether the endospermic ABA contributes to thermoinhibition is not known[2].

In a seed coat bedding assay (SCBA), embryos are cultured on a bed of dissected endosperms (with the seed coat still attached) to study the influence of the endosperm on embryo growth[16]. WT embryos cultured on a bed of WT endosperms under high temperatures were unable to grow, maintaining the same appearance as embryos within intact seeds, and accumulated high levels of the ABA response transcription factor ABI5 (Fig. 1a, b, d, e)[19]. Furthermore, WT embryos cultured on a bed of *aba1* endosperms, unable to synthesize ABA, were able to grow and had low ABI5 accumulation (Fig. 1b, d, e). These observations strongly indicated that seed thermoinhibition involves ABA release by the endosperm towards the embryo to block its growth.

Consistent with this view, endosperms cultured at 30 °C for 48 h contained 8-fold more ABA than those cultured at 22 °C (Fig. 1f). Furthermore, over the same period, the levels of ABA in the culture medium at 30 °C were more than 16x higher than those at 22 °C (Fig. 1f). These observations show that cultured dissected endosperms actively synthesize and release ABA at 30 °C but not at 22 °C. These results support the notion that the endosperm is necessary to implement thermoinhibition by synthesizing and releasing ABA towards the embryo. Interestingly, *aba1* mutant embryos were able to grow when cultured on a bed of WT endosperms and did not accumulate ABI5 (Fig. 1b, d, e). This indicates that ABA synthesized in the embryo also participates to promote thermoinhibition without being sufficient. To further asses this notion, we examined the embryonic expression of ABA biosynthesis genes (*NCED2* and *NCED9*) and of an ABA catabolic gene (*CYP707A1*). Consistent with our hypothesis, we found that *NCEDs* and *CYP707A1* expression markedly increase and decrease, respectively, under high temperatures relative to normal temperatures (Supplementary Fig. 1a).

### Endospermic phyB signaling mediates seed thermoinhibition
Previous work showed that the endosperm blocks the germination of non-dormant seeds in response to FR light. Indeed, in the endosperm FR light converts the phytochrome B (phyB) Pfr signaling active form into its Pr inactive form, which promotes ABA accumulation and release towards the embryo to block its germination[14]. Phytochrome signaling in seedlings was shown to be affected by temperature. Indeed, increasing temperatures promotes Pfr reversion into the Pr inactive form and promotes hypocotyl growth[12,13]. We therefore hypothesized that high temperatures block germination by repressing phyB signaling in the endosperm.

In the Col-0 accession, seed thermoinhibition in six months-old seeds is weak at 28 °C and is better observed at temperatures in the 30–34 °C range (Fig. 2a)(see discussion). At 34 °C, seed germination was blocked in WT, *phyB* and *phyA phyC phyD phyE* (*phyACDE*) mutant seeds (Fig. 2a). In contrast, at 28 °C WT and *phyACDE* seeds germinated whereas *phyB* mutants did not germinate, consistent with a previous report (Fig. 2a)[20]. At 22 °C, *phyB* and *phyACDE* mutant seeds germinated with the same percentage as WT seeds. Hence, seeds deficient in phyB signaling maintain seed thermoinhibition when temperature drops from 34 °C to 28 °C, unlike WT seeds. This suggests that absence of seed thermoinhibition at 28 °C in WT seeds requires active phyB signaling and, conversely, that thermoinhibition at 34 °C in WT seeds

could be due to inactive phyB signaling. At 22 °C, *phyB* mutants germinated, suggesting that other phytochromes promote germination at lower temperatures (see discussion).

Strikingly, at 28 °C *phyB* embryos deprived of their endosperm were able to grow, even reaching a longer hypocotyl length relative to WT embryos after 4 days, despite the strong thermoinhibition observed in intact *phyB* mutant seeds (Fig. 2b). The longer hypocotyl of *phyB* mutant seedlings arising from endosperm-less embryos is consistent with previous work showing that phyB signaling represses hypocotyl elongation in seedlings[11]. Furthermore, WT embryos cultured on a bed of *phyB* mutant endosperm at 28 °C had their growth strongly repressed in comparison to that of WT embryos cultured on a bed of WT endosperms and accumulated higher ABI5 protein levels (Fig. 2c, d). At 22 °C, WT and *phyB* endosperms were unable to block the growth of WT embryos nor to trigger high ABI5 accumulation (Fig. 2c, d). Hence, these observations strongly indicate that *phyB* seed thermoinhibition at 28 °C is implemented by higher endospermic ABA release relative to WT endosperm due to lack of phyB signaling.

Previous reports have shown that phyB thermal reversion can be accelerated or slowed down by specific amino acid changes. We compared a *phyB* mutant complementation line in which thermal reversion is accelerated by a Ser86Asp amino acid change (phyB^{S86D}) with a control line bearing the WT phyB sequence (phyB)(Methods)[21]. At 30 °C, the phyB^{S86D} line had higher thermoinhibition than phyB even though both lines accumulated similar amounts of transgenic gene products (Fig. 2e). Stronger thermoinhibition in the phyB^{S86D} line is most likely the result of stronger endosperm-imposed germination arrest because the growth of phyB^{S86D} embryos was faster than that of phyB embryos upon endosperm removal (Fig. 2e). We also assessed thermoinhibition in *phyB* mutant transgenic lines in which phyB thermal reversion is slowed down by a G564E amino acid change (phyB^{G564E}) relative to a control line bearing no change (phyB)[22,23]. At 32 °C the phyB^{G564E} line was less thermoinhibited than the phyB control line even though the hypocotyl of the control line elongated faster than that of the phyB^{G564E} line (Fig. 2f). The complemented lines accumulated similar amounts of transgenic gene products at 32 °C (Fig. 2f). Altogether, these results support the notion that high temperatures promote endospermic phyB thermal reversion, which leads to higher endospermic ABA levels and release towards the embryo to block its growth.

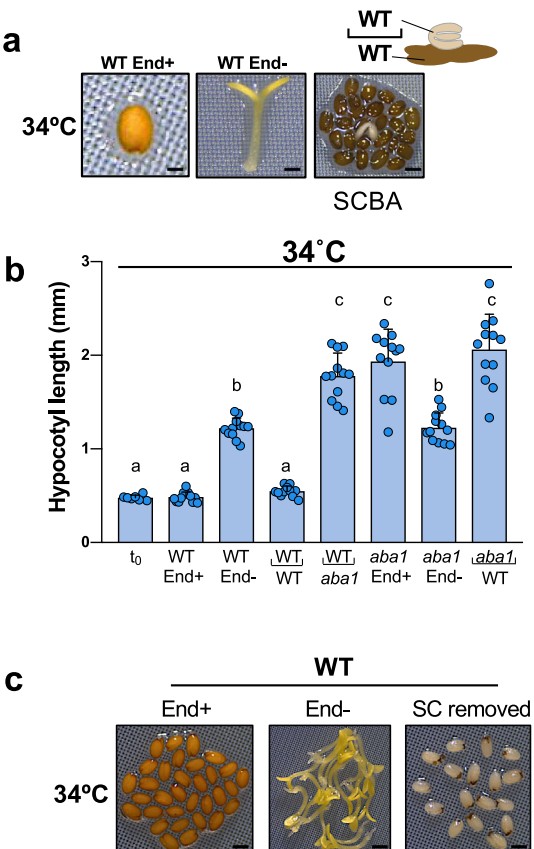

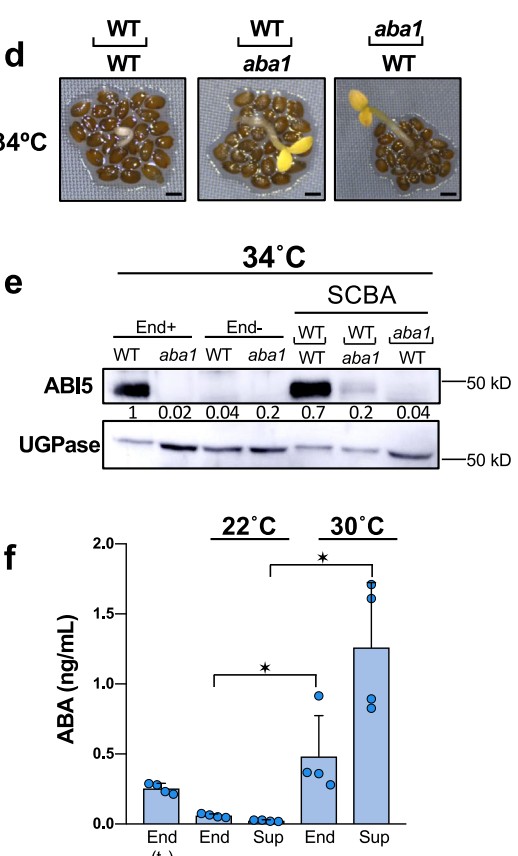

**Fig. 1 | The endosperm enables seed thermoinhibition by releasing ABA. a** Left (WT End+): WT seed 3 days after imbibition at 34 °C (bar: 0.1 mm). Middle (WT End−): same experiment with the endosperm removed 4 h after seed imbibition (bar: 0.3 mm). Right (SCBA): same experiment with WT embryo cultured on a bed of WT endosperms (bar: 0.5 mm). **b** Hypocotyl length (mm) of WT and *aba1* embryos (*n* = 12) cultivated as in **a**. The hypocotyl length of a WT embryo dissected 4 h after seed imbibition is provided for reference (t₀). Lower case letters above histograms are used to establish whether two values are statistically significantly different as assessed by one-way ANOVA followed by a Tukey HSD test (*p* < 0.05): different letters denote statistically different values. **c** Left (End+): WT seeds as in **a**. (left)(bar: 0.5 mm). Middle (End−): WT embryos as in **a**. (middle) (bar: 1 mm). Right (SC removed): WT seeds cultivated as in **a**. with the seed coat removed 4 h after seed imbibition (bar: 0.5 mm). **d** SCBAs using WT and *aba1* embryos cultivated on a bed of WT or *aba1* endosperms as in a. Bar: 0.5 mm (left, center) and 0.8 mm (right). Hypocotyl length measurements are shown in **b**. **e** ABI5 protein levels in WT and *aba1* embryos dissected from seeds 3 days upon seed imbibition (End+) and in those dissected 4 h after seed imbibition and cultured for 3 days (End−) at 34 °C. For the SCBAs, WT or *aba1* embryos were cultured on a bed of WT or *aba1* endosperms as indicated. Numbers represent ABI5 levels normalized to those of UGPase. **f** WT endosperms were dissected 4 h after seed imbibition and cultured in water at 22 °C or 30 °C for 40 h. Histograms show ABA levels measured in the endosperm tissue at the time of dissection (t₀) and 40 h after incubation (End) and measured in culture medium (Sup). Data from 4 independent experiments (500 endosperms per experiment). One-way ANOVA followed by a Tukey HSD test shows statistically different values (*p* < 0.05). All plant material in Fig. 1 is cultivated under white light (50 μmol/m²/s). Errors bars represent standard deviations.

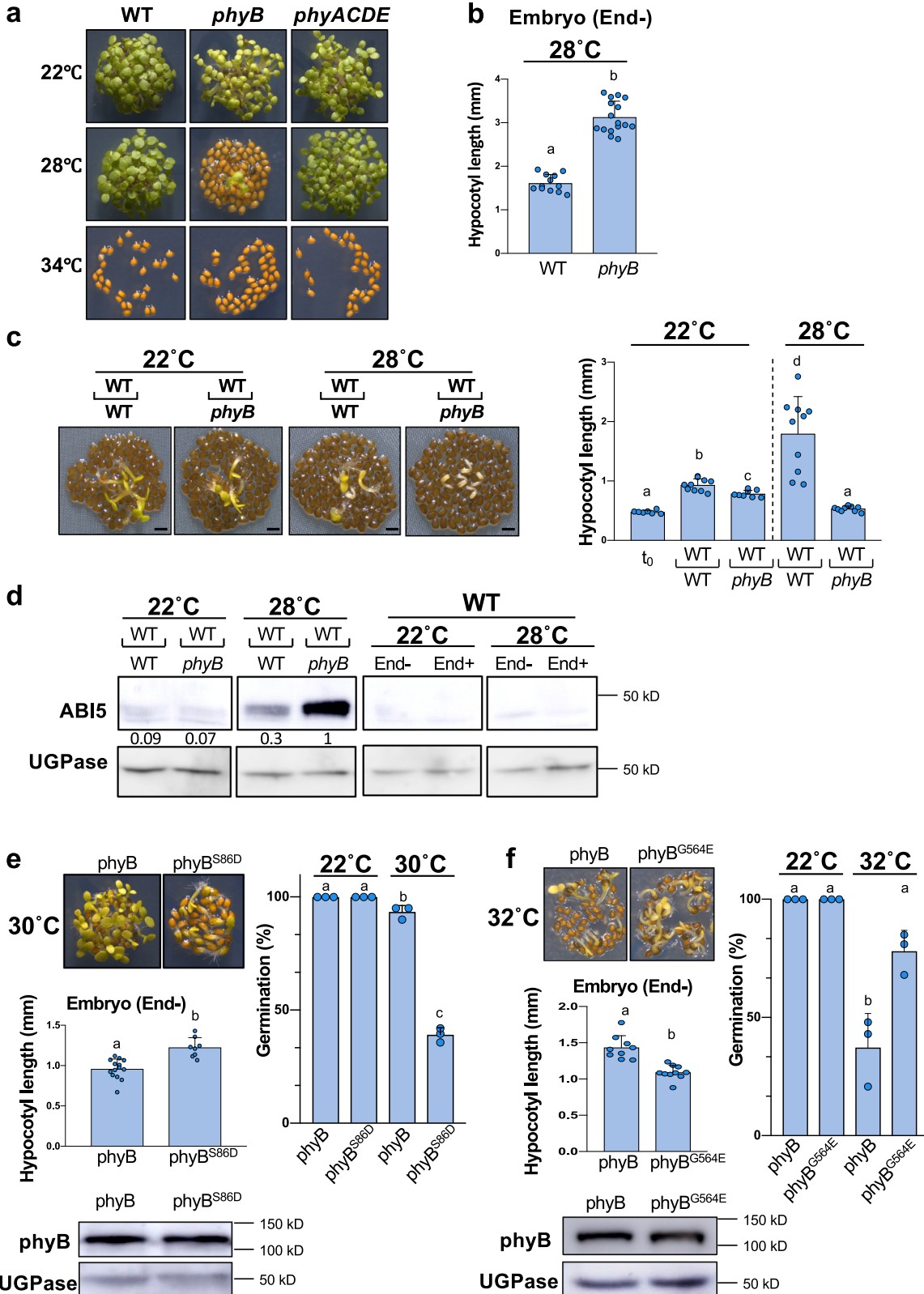

## DELLAs promote endospermic ABA release under high temperatures

We next sought to identify the endospermic factors promoting ABA accumulation and release when phyB signaling is repressed by high temperatures. The DELLA factors were potential candidates for several reasons: 1) low phyB signaling represses the expression of GA biosynthesis genes, 2) the DELLA factors RGL2, GAI and RGA promote ABA accumulation when GA levels are low and 3) *rgl2* mutants seeds have low seed thermoinhibition responses[2,6,7,24]. We explored whether endospermic DELLA factors promote thermoinhibition. At 32 °C, *della* mutants seeds, bearing loss-of-function mutations in all five *Arabidopsis* DELLA genes (*RGL2, GAI, RGA, RGL1* and *RGL3*), lacked seed thermoinhibition unlike WT seeds (Supplementary Fig. 1b). Furthermore, at 32 °C, *della* endosperms did not repress WT embryo growth

**Fig. 2 | Endospermic phyB signaling mediates seed thermoinhibition. a** WT, *phyB* and *phyACDE* plants 4 days after seed imbibition at 22 °C, 28 °C or 34 °C. **b** Hypocotyl length (mm) of WT and *phyB* embryos dissected 4 h after seed imbibition and cultured for 4 days at 28 °C ($n = 12$ and $n = 16$ for WT and *phyB*, respectively). Statistical treatment and lower-case letters as in Fig. 1b. **c** Left panel: SCBAs using WT embryos cultivated on a bed of WT or *phyB* endosperms as in Fig. 1a using 22 °C and 28 °C. Bar: 0.8 mm. Right panel: WT hypocotyl lengths (mm) are measured at the time of dissection ($t_0$) and after 3 days ($n = 7$ for $t_0$, $n = 8$ for WT/ *phyB* SCBA and $n = 10$ for the rest). Statistical treatment and lower-case letters as in Fig. 1b. **d** First two panels: ABI5 protein levels 3 days upon seed imbibition at 22 °C or 28 °C in WT embryos cultured on a bed of WT or *phyB* endosperms as indicated. Last two panels: ABI5 levels in embryos 3 days upon seed imbibition at 22 °C or 28 °C in WT seeds (End+) and embryos upon removal of the endosperm 4 h after

seed imbibition (End−). Numbers represent ABI5 levels normalized to those of UGPase. **e** *phyB* mutant seeds complemented with a transgene expressing WT phyB (phyB) or a modified phyB with Ser86Asp (phyB$^{S86D}$) incubated for 3 days at 30 °C. Average germination percentage after 3 days at 22 °C and 30 °C in three independent seed batches ($n \geq 50$). Statistical treatment and lower-case letters as in Fig. 1b. Hypocotyl length (mm) of embryos dissected 4 h after seed imbibition and cultured for 3 days at 30 °C ($n = 8$ for phyB$^{S86D}$ and $n = 14$ for phyB). Statistical treatment and lower-case letters as in Fig. 1b. phyB protein levels in seeds of the two phyB lines incubated for 24 h at 30 °C. UGPase protein levels were used as a loading control. **f** Same as e using *phyB* mutant seeds complemented with a transgene expressing WT phyB (phyB) or a modified phyB with Gly564Glu (phyB$^{G564E}$). All plant material in Fig. 2 cultivated under white light (50 μmol/m²/s). Errors bars represent standard deviations.

nor induce high embryonic ABI5 accumulation in a SCBA unlike WT endosperms (Fig. 3a). These observations strongly suggest that endospermic DELLA factors promote thermoinhibition by enhancing endospermic ABA levels and release under high temperatures.

We next evaluated the hypothesis that DELLA factors promote endospermic ABA when high temperatures reduce endospermic phyB signaling. phyB signaling promotes synthesis of GA, which binds to the GID1 receptors leading to their interaction with DELLA factors and promotion of DELLA proteasomal degradation[25–28]. At 22 °C, RGL2 protein levels were similar in the endosperm and embryo of WT and *phyB* seeds (Fig. 3b). At 28 °C, a temperature that blocks the germination of *phyB* seeds but not that of WT seeds, RGL2 levels marginally increased in the endosperm and embryo of WT seeds (Fig. 3b). In contrast, at 28 °C RGL2 levels strongly increased in both tissues of *phyB* mutant seeds (Fig. 3b). At 32 °C, a temperature blocking WT seed germination, RGL2 levels strongly increased in the endosperm and embryo of WT seeds (Fig. 3c).

At high temperatures, phyB accumulation was moderately lower in *della* mutant seeds relative to WT seeds indicating that enhanced phyB signaling that would result from high phyB levels in *della* seeds does not account for their lack of thermoinhibition (Supplementary Fig. 1c). *della* mutant seeds irradiated with FR light upon imbibition, which inactivates phyB, and maintained at 30 °C in darkness continued to lack seed thermoinhibition, unlike WT seeds (Fig. 3d). Consistent with this observation, at 30 °C, FR-irradiated *della* endosperm released 1.5-fold lower ABA levels relative to WT endosperm (Fig. 3e). In addition, under white light, *della phyB* mutant seeds, lacking DELLA factors and phyB, were not thermoinhibited at 28 °C, unlike *phyB* mutant seeds (Supplementary Fig. 1d). Altogether, these observations indicate that DELLA factors promote thermoinhibition when phyB signaling is repressed by high temperatures.

### Endospermic ABA release without DELLAs under high temperatures

Interestingly, however, several observations suggested that upon lowering of phyB signaling by high temperatures another endospermic pathway promotes ABA accumulation and release independently of DELLA factors. Indeed, 1) thermoinhibition could be observed at 30 °C in about a quarter of *della* seeds irradiated with FR, which inactivates phyB and, furthermore, the percentage of thermoinhibited *della* seeds further increased with increasing temperatures, which accelerates phyB reversion, reaching 100% at 34 °C (Fig. 3d); 2) *della phyB* mutant seeds, lacking DELLA factors and phyB, were more thermoinhibited at 30 °C than *della* mutant seeds (Supplementary Fig. 1e); 3) At 30 °C, FR-irradiated *della* endosperms continued to release substantial amounts ABA even though they were lower than those released by FR-irradiated WT endosperms (Fig. 3e); 4) at 34 °C in presence of white light, a substantial percentage (40%) of *della* seeds were thermoinhibited whereas no thermoinhibition took place in *aba1* seeds, deficient in ABA synthesis

(Supplementary Fig. 1b) and 5) *della* seed thermoinhibition required the presence of the endosperm and, accordingly, at 34 °C, *della* endosperms were able to repress the growth of WT embryos and induce ABI5 accumulation in a SCBA (Fig. 3f). Finally, Toh et al observed that WT seed thermoinhibition at 34 °C was not fully abolished in presence of exogenous GA, which promotes DELLA factor degradation[2]. We therefore sought to identify additional endospermic factors promoting ABA accumulation and release.

### Endospermic PIF3 promotes ABA release under high temperatures

In its active Pfr form phyB interacts with PIFs transcription factors to promote their degradation so that PIFs are stabilized when the levels of the phyB Pfr form are low, which promotes PIFs accumulation[10,11]. We therefore asked whether PIFs could promote seed thermoinhibition. *pifq7* mutant seeds, lacking PIF1, PIF3, PIF4, PIF5, PIF7, were thermoinhibited at 32 °C, indicating that PIFs are not essential for seed thermoinhibition at very high temperatures (Supplementary Fig. 2a)[29]. We therefore monitored seed thermoinhibition responses at 30 °C, a moderately high temperature that does not fully promote thermoinhibition in WT seeds (Fig. 4a). At 22 °C, no noticeable differences in seed germination were observed between WT and various single or multiple *pif* mutant seeds (Supplementary Fig. 2b). At 30 °C, *pif4*, *pif6* and *pif7* seed thermoinhibition was similar to that of WT seeds. *pif1* seed thermoinhibition was lower than WT in some seed batches (Batches 1 and 2) but not in others (Batches 3 and 4)(Fig. 4a). *pif5* thermoinhibition was lower in two seed batches whereas *pif3* thermoinhibition was lower in four seed batches (Fig. 4a). Combining the *pif3* mutation with *pif1* and/or *pif5* mutations lead to even lower thermoinhibition in all seed batches (Fig. 4a). The double mutant *pif3 pif4* was also less thermoinhibited than *pif3* suggesting that PIF4 could also promote thermoinhibition in absence of PIF3. At higher temperatures (32 °C), these same *pif* mutant seeds increased their thermoinhibition, suggesting that DELLA factors overcome the absence of PIF1, PIF3 and PIF5 to promote thermoinhibition (Supplementary Fig. 2c). Consistent with this hypothesis, *pif1*, *pif5* and *pif3* mutant thermoinhibition in presence of GA (5 μM) strongly decreased relative to WT at 32 °C and even further decreased in the double and triple mutant combinations (Supplementary Fig. 2c). Altogether, these data show that PIFs, and notably PIF1, PIF3 and PIF5, promote seed thermoinhibition. Together with the observation that *pifq7* mutants are thermoinhibited at very high temperatures (Supplementary Fig. 2a), our results indicate that DELLA factors and PIFs promote endospermic ABA accumulation and release through parallel pathways.

PIFs were previously shown to promote hypocotyl elongation, which contrasts with their activity to promote seed thermoinhibition, which blocks the embryo-to-seedling transition[10,11,30]. This is consistent with the notion that PIFs-mediated thermoinhibition activity takes place in the endosperm, rather than in the embryo, by promoting ABA accumulation and release.

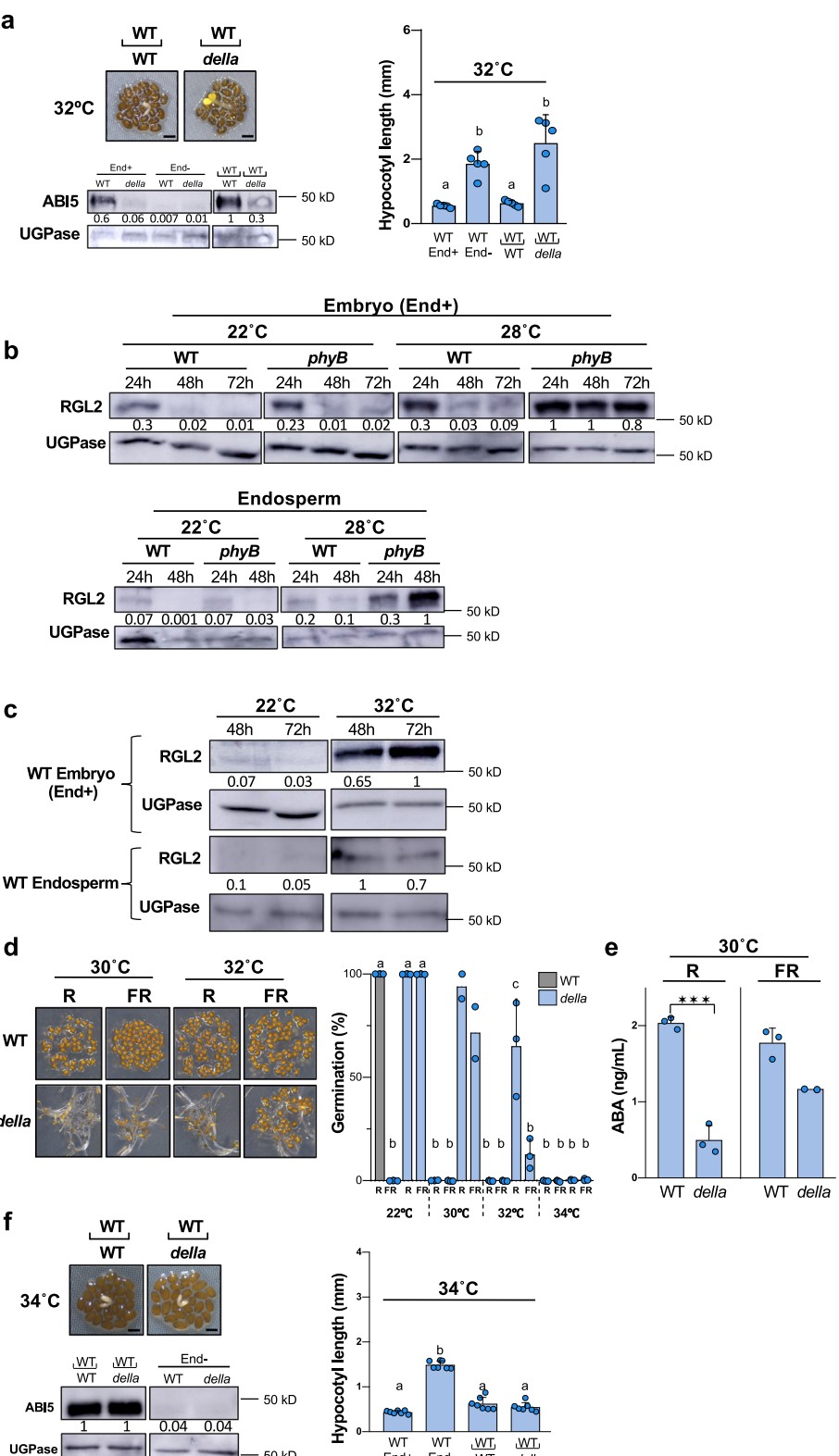

We first sought to assess embryonic PIFs activity to promote early seedling growth in embryos deprived of their endosperm. We measured the hypocotyl length of 3-day-old WT and different *pif* mutant seedlings derived from embryos whose endosperm was removed 4 h upon seed imbibition (Fig. 4b). Hypocotyls elongated faster at 30 °C and 32 °C than at 22 °C in all genotypes tested, consistent with previous reports (Fig. 4b)[8,31]. Interestingly, relative to WT, the hypocotyl of

the *pif3* mutant was markedly shorter at 30 °C (34% shorter) and 32 °C (32% shorter) and to a lesser extent at 22 °C (20% shorter) whereas that of the other *pif* mutants was similar with the exception of *pif4*, which was slightly shorter at 22 °C (12% shorter) and 32 °C (14% shorter) (Fig. 4b). Koini et al. examined hypocotyl elongation of 4-day-old seedlings upon transfer to high temperatures (28 °C) and found that the hypocotyl of *pif4* seedlings elongated less than WT, whereas that of

**Fig. 3 | DELLAs promote endospermic ABA release under high temperatures.**
**a** Left panel: SCBAs using WT embryos cultivated on a bed of WT or *della* endosperms as in Fig. 1a using 32 °C. Bar: 0.5 mm. Hypocotyl lengths (*n* = 5) (Right panel) and ABI5 protein levels (Bottom panel) in the WT or *della* embryos dissected 4 h after seed imbition and cultured for 3 days at 32 °C without endosperm (End−), in the embryos dissected after high temperature seed incubation (4 h at 22 °C +3 days at 32 °C) (Endo+), and in the embryo dissected and cultured for 3 days at 32 °C on a bed of WT or *della* endosperms. Numbers represent ABI5 levels normalized to those of UGPase. Statistical treatment and lower-case letters are used as in Fig. 1b.
**b** RGL2 protein levels in WT and *phyB* embryos (upper panel) and endosperms (lower panel) from seeds cultivated at 22 °C and 28 °C and dissected at the indicated times upon seed imbition. Numbers represent RGL2 levels normalized to those of UGPase. **c** Same as 3b using WT seeds cultivated at 22 °C and 32 °C. All plant material in **a**–**c** cultivated under white light (50 μmol/m²/s). **d** WT and *della*

seeds cultivated in darkness for 3 days at 30 °C or 32 °C after receiving a 5 min far-red light (FR) pulse (3.69 μmol/m²/s) or a FR pulse followed by a 5 min red light (R) pulse (14.92 μmol/m²/s) 2 h upon seed imbition. Average germination percentages after 3 days at indicated temperatures in 3 independent seed batches (2 for 30 °C) (*n* ≥ 50). Statistical treatment and lower-case letters are used as in Fig. 1b. **e** WT and *della* endosperms dissected 4 h after seed imbition were suspended in water and irradiated with FR/R or FR pulses (as in **d**) and then cultured in darkness at 30 °C for 40 h. Histograms show ABA levels measured in culture medium after 40 h. Data from 3 biological samples (2 for *della* at 30 °C) (150 endosperms per experiment). Statistical treatment is used as in Fig. 1f (***$p < 0.001$). **f** Same as Fig. 3a but seeds were incubated at 34 °C and seven hypocotyl lengths were measured (*n* = 7). Statistical treatment and lower-case letters are used as in Fig. 1b. Errors bars represent standard deviations.

*pif3* seedlings responded normally[8]. We could confirm these observations and, together with our observations, we conclude that PIF3 plays a specific role to promote early hypocotyl elongation particularly at high temperatures (Supplementary Fig. 2d). This conclusion, together with the fact that PIF3 was not previously involved in temperature responses nor in the control of seed germination (unlike PIF1), prompted us to further characterize PIF3 function during thermoinhibition.

We assessed PIF3 activity in the endosperm to promote ABA accumulation and release. We dissected WT and *pif3* endosperms 4 h after seed imbition and cultured them in water at 30 °C for 48 h. The *pif3* culture medium contained significantly lower ABA levels than that of the WT culture medium (Fig. 4c). This result shows that PIF3 activity in endosperm is necessary to release ABA at high temperatures. Consistent with this conclusion, *pif3 phyB* seeds were less thermoinhibited than *phyB* seeds at 28 °C (Fig. 4d). Moreover, the *della pif3* mutant was less thermoinhibited at 30 °C and 32 °C than the *della* quintuple mutant showing that PIF3 can promote thermoinhibition independently of DELLA factors (Fig. 4e)(Methods). Altogether, these results corroborate the view that PIF3 promotes endospermic ABA accumulation and release when phyB signaling is repressed by high temperatures. Furthermore, these findings also lead us to conclude that in absence of the endosperm, PIF3 indeed promotes early seedling growth under high temperatures. This conclusion is rather intriguing since on one hand, PIF3 promotes embryonic growth in absence of the endosperm but on the other it represses growth in presence of the endosperm. To better understand this apparent paradox, we sought to understand how PIF3 protein accumulation is regulated under high temperatures in the endosperm and embryo.

### Endospermic ABA represses embryonic *PIF3* expression
Remarkably, in WT seeds imbibed at 22 °C in presence of white light PIF3 protein accumulation increased markedly but transiently at 36 h and 48 h in the embryo upon seed imbition, which is the time when the embryo is germinating, and the hypocotyl of the seedling being formed rapidly elongates (Fig. 5a). This is consistent with the role of PIF3 to promote early seedling growth as shown above. In contrast, at 34 °C, PIF3 embryonic accumulation remained low over time upon imbition (Fig. 5a). However, endospermic PIF3 accumulation was lower at 22 °C than at 34 °C (Fig. 5a). A transgenic line carrying a *PIF3* promoter sequence fused to GUS (prom*PIF3::GUS*) yielded results consistent with those of PIF3 accumulation: at 22 °C, embryonic GUS activity increased strongly between 24 h and 48 h upon imbition and diminished thereafter whereas it was undetected in thermoinhibited embryos at 34 °C (Fig. 5b). Conversely, endosperm GUS activity was higher at 34 °C than at 22 °C (Fig. 5b). This suggests that at 22 °C the observed increases in embryonic PIF3 accumulation are driven by increased embryonic *PIF3* mRNA accumulation whereas at 34 °C, low embryonic PIF3 levels are due to low embryonic *PIF3* mRNA levels. In contrast to the embryo, *PIF3* mRNA accumulation in the endosperm is

stimulated by high temperatures, which promotes endospermic PIF3 accumulation.

In absence of endosperm, embryos cultured at 22 °C and 34 °C similarly accumulated transiently PIF3 (Fig. 5c). This suggested that ABA released by the endosperm represses PIF3 accumulation in the embryo. Consistent with this notion, embryos cultured in presence of exogenous ABA maintained low PIF3 levels over time at both 22 °C and 34 °C (Fig. 5c). Furthermore, exogenous ABA also strongly reduced GUS activity in prom*PIF3::GUS* transgenic embryos (Fig. 5b). In addition, 48 h upon imbition at 28 °C, a temperature where *phyB* seeds are thermoinhibited, *phyB* embryos surrounded by the endosperm accumulated markedly lower PIF3 levels than embryos deprived of their endosperm (Fig. 5d). No such differences in PIF3 accumulation were observed at 22 °C (Fig. 5d). Altogether, these results show that under high temperatures endospermic ABA inhibits the accumulation of *PIF3* mRNA and PIF3 protein in the embryo.

### PIF3 regulates different genes in the endosperm and embryo
To identify the genes regulated by PIF3 under higher temperatures in the endosperm and embryo, we performed an RNA-seq analysis. We compared the transcriptome of WT and *pif3* endosperm from seeds imbibed for 24 h and 48 h at 30 °C, which is a time when WT and *pif3* seeds have not yet germinated (Supplementary Fig. 3). We also compared the transcriptome of WT and *pif3* embryos, after removing their endosperm 4 h upon seed imbition, cultured for 24 h and 48 h at 30 °C, which are times when WT and *pif3* hypocotyl length was comparable (Supplementary Fig. 3).

At 24 h, we found that 73 and 25 genes were up-regulated and down-regulated, respectively, in *pif3* endosperms relative to WT endosperms whereas 577 and 228 were up-regulated and down-regulated, respectively, in *pif3* embryos relative to WT embryos (Supplementary Fig. 4a and Data 1). Among those, only 23 genes were misregulated in both *pif3* endosperm and *pif3* embryo (15 upregulated and 8 downregulated). At 48 h, as many as 1179 and 1191 genes were up-regulated and down-regulated, respectively, in *pif3* endosperms relative to WT endosperms whereas only 130 and 186 were up-regulated and down-regulated, respectively, in *pif3* embryos relative to WT embryos (Supplementary Fig. 4a and Data 1). Among those, only 16 genes were misregulated in both *pif3* endosperm and *pif3* embryo (10 upregulated and 6 downregulated)(Supplementary Fig. 4a and Data 1). We performed a gene ontology (GO) enrichment analysis and observed that for the upregulated genes at 24 h olefinic compound metabolic process-related genes are highly enriched in the endosperm while photosynthesis-related and water transport-related genes are highly enriched in the embryo (Supplementary Fig. 4b). For the downregulated genes at 24 h, response to heat-related genes are highly enriched in the endosperm while seed oil body biogenesis- and lipid storage-related genes are highly enriched in the embryo (Supplementary Fig. 4b). For the upregulated genes at 48 h, callose deposition-, cell wall thickening-, and polysaccharide localization-

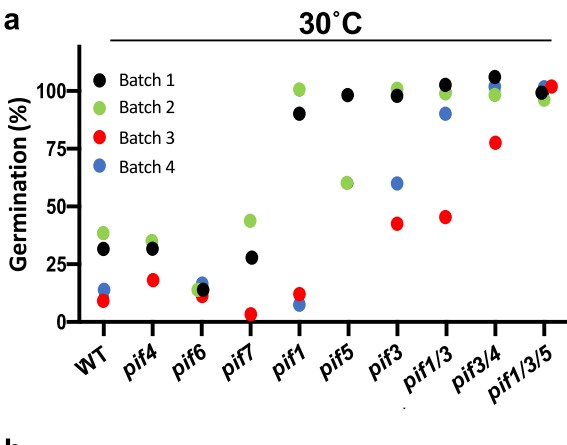

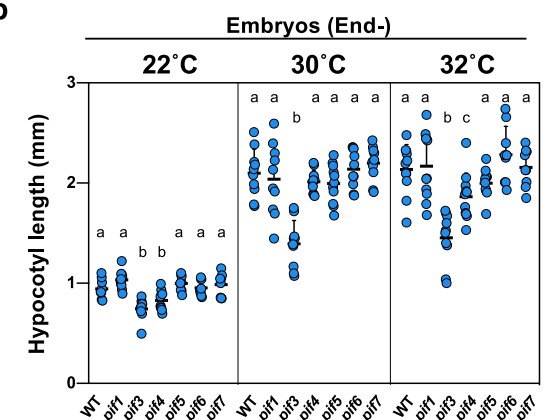

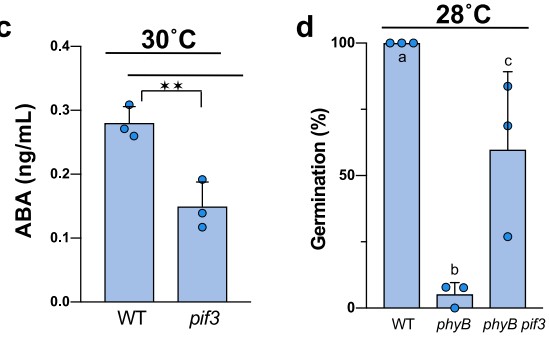

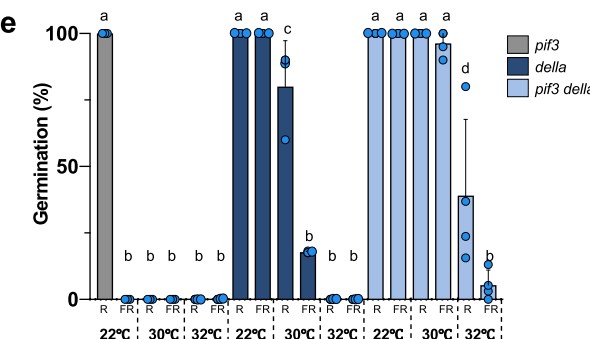

**Fig. 4 | Endospermic PIF3 promotes ABA release under high temperatures.**
**a** Average germination percentages of WT seeds and seeds carrying mutations in *PIF* genes, as indicated, after 4 days under white light (50 μmol/m²/s) at 30 °C in 4 independent seed batches (*n* ≥ 50 seeds). **b** Hypocotyl length (mm) of WT and *pif* embryos dissected 4 h after seed imbibition and cultured for 3 days under white light (50 μmol/m²/s) at 22 °C, 30 °C and 32 °C (*n* = 10). Statistical treatment and lower-case letters are used as in Fig. 1b. **c** WT and *pif3* endosperms were dissected 4 h after seed imbibition from WT and *pif3* seeds and cultured in water at 30 °C under white light (50 μmol/m²/s) for 40 h. Histograms show ABA levels measured in culture medium after 40 h. Data from 3 independent experiments (150 endosperms per experiment). One-way ANOVA followed by a Tukey HSD test shows statistically different values (**$p < 0.01$). **d** Average germination percentages of WT, *phyB* and *phyB pif3* seeds after 3 days under white light (50 μmol/m²/s) at 28 °C in 3 independent seed batches (*n* ≥ 50 seeds). Statistical treatment and lower-case letters are used as in Fig. 1b. **e** Average germination percentages of *pif3*, *della* and *pif3 della* seeds cultivated in darkness for 3 days at 22 °C, 30 °C and 32 °C after receiving a 5 min far-red light (FR) pulse (3.69 μmol/m²/s) or a FR pulse followed by 5 min red light (R) pulse (14.92 μmol/m²/s) 2 h upon seed imbibition in 3 independent seed batches (4 for *pif3 della*)(*n* ≥ 50 seeds). Statistical treatment and lower-case letters are used as in Fig. 1b. Errors bars represent standard deviations.

representations of gene expression show that different sets of genes are regulated by PIF3 in the endosperm and the embryo at both 24 h and 48 h (Supplementary Fig. 4c).

Altogether, these data are consistent with the notion that endospermic PIF3 could play distinct gene regulatory roles relative to those of embryonic PIF3.

Among ABA and GA metabolism genes, only the ABA biosynthesis gene *NINE-CIS-EPOXYCAROTENOID DIOXYGENASE 3* (*NCED3*) and the ABA catabolic gene *CYP707A1*, encoding an ABA 8'- hydroxylase, were consistently upregulated in the *pif3* endosperm relative to the WT endosperm at 24 h and 48 h (Supplementary Fig. 4c and Data 1). Given that we found that *pif3* endosperm produces less ABA compared with WT at high temperatures (Fig. 4c), we further examined the role of *CYP707A1* in thermoinhibition. We confirmed by qPCR that *CYP707A1* is upregulated in *pif3* endosperm relative to WT 48 h upon seed imbibition (Supplementary Fig. 5a). We found that 2-month-old WT, *pif3*, *cyp707a1* and *pif3 cyp707a1* seeds fully germinated 3 days upon imbibition at 22 °C (Supplementary Fig. 5b). In contrast, *cyp707a1* were fully thermoinhibited at 28 °C, unlike WT seeds, strongly suggesting that CYP707A1 plays a significant role to regulate thermoinhibition, likely by limiting ABA levels in seeds (Supplementary Fig. 5b). Furthermore, *pif3 cyp707a1* seeds were also fully thermoinhibited at 28 °C (Supplementary Fig. 5b). Given that *pif3* seeds are less thermoinhibited at 30 °C relative to WT (Fig. 4a), these results show that *cyp707a1* is epistatic to *pif3*, suggesting that CYP707A1 functions downstream of PIF3 in the endosperm (Fig. 6).

## Discussion

Here we showed that the endosperm is essential for seed thermoinhibition to take place in *Arabidopsis thaliana*. Our results support the model that high temperatures lower endospermic phyB signaling to promote endospermic ABA accumulation and release via two parallel signaling branches: one involving the DELLA factors and another involving the PIFs factors, mainly PIF1, PIF3 and PIF5 (Fig. 6). This model does not exclude that each branch could also regulate the activity of the other. Indeed, previous work using whole seeds showed that PIF1 represses and promotes GA synthesis and *DELLA* gene expression, respectively[33,34]. Thus, other PIFs could similarly influence GA signaling responses in the endosperm under high temperatures. Furthermore, DELLA and PIFs factors are known to interact with each other, modulating each other's stability and activity[10,35].

Previous work showed that DELLA factors promote ABA accumulation, which we further confirmed here in the context of seed thermoinhibition[6]. However, we provided evidence that PIFs also

related genes are highly enriched in the endosperm while response to cold-related genes are enriched in embryo. For the downregulated genes at 48 h, various RNA processing or metabolic process related genes are highly enriched in endosperm while no specific genes are enriched in embryo (Supplementary Fig. 4b). We considered genes bound by PIF3[32], which are potential direct targets of PIF3, genes involved in ABA and GA metabolism and signaling to analyze their expression changes in the *pif3* endosperm and embryo. Heatmap

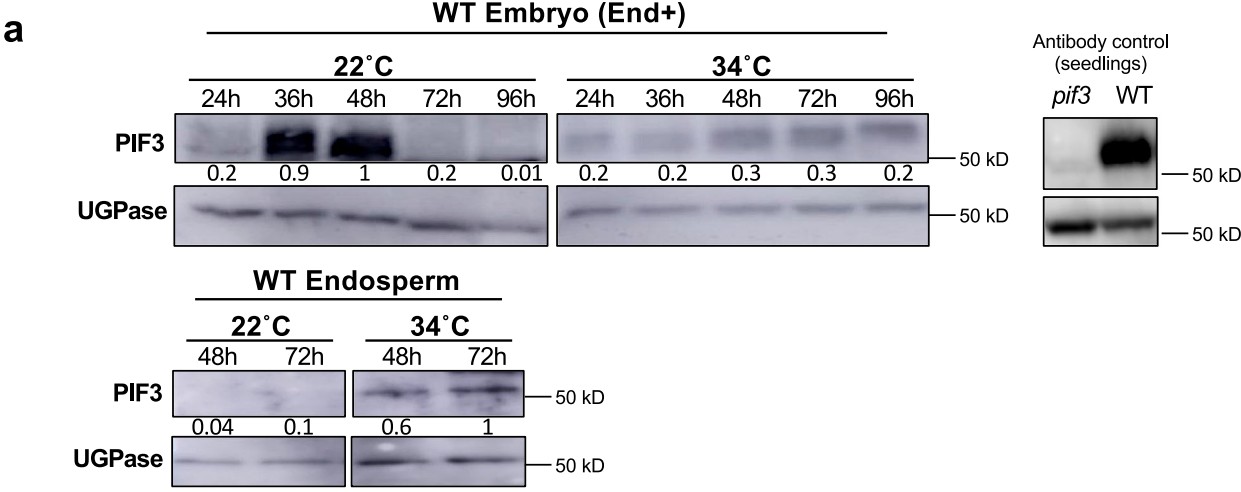

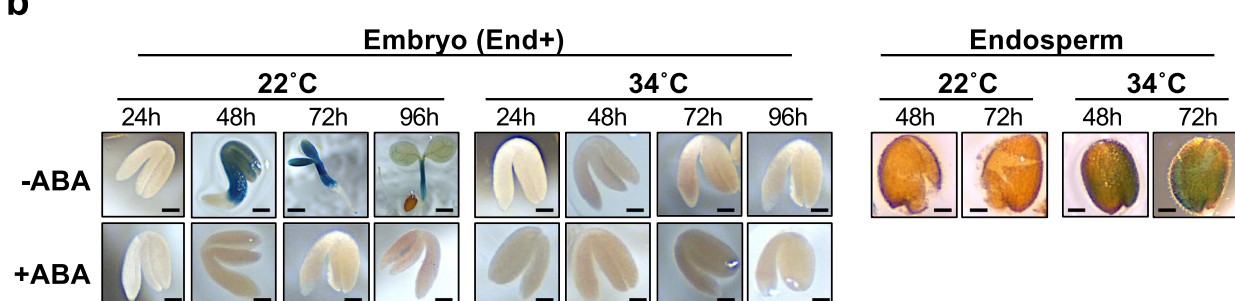

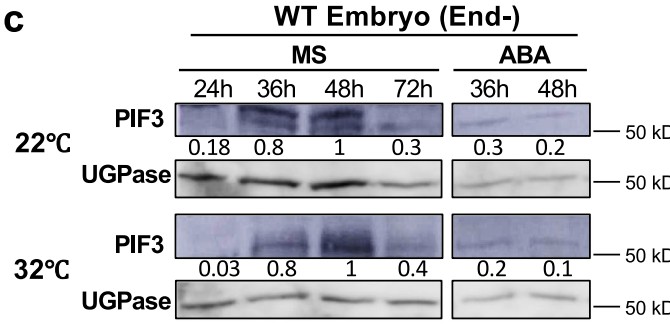

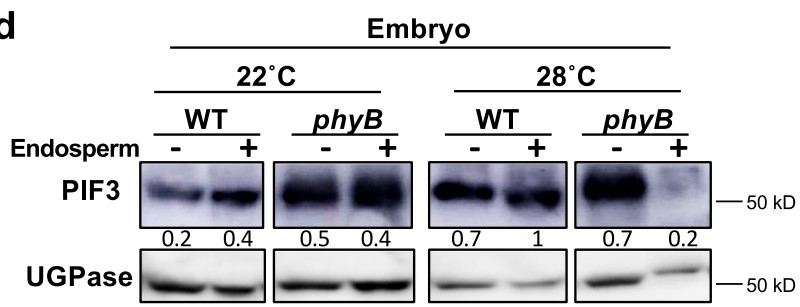

promote ABA accumulation and thermoinhibition independently of DELLA factors. In the case of PIF3, this involves PIF3-mediated repression of *CYP707A1* expression. Furthermore, PIF1 was shown to promote ABA accumulation by promoting expression of the ABA biosynthesis genes *NINE-CIS-EPOXYCAROTENOID DIOXYGENASE6* (*NCED6*) and *NCED9* and repressing that of *CYP707A2*, encoding another important catabolic ABA 8′-hydroxylase in seeds[18,33].

Previous work has shown that PIF4 mediates developmental responses to temperature such as promoting hypocotyl and petiole growth in seedlings or flowering in response to high temperatures[8,9]. Concerning seed thermoinhibition, we found that PIF4 plays a small role, if any, to promote seed thermoinhibition relative to PIF1, PIF3 and PIF5 but its contribution to thermoinhibition can be observed in a *pif3* mutant background (Fig. 4a).

**Fig. 5 | PIF3 activity in the endosperm represses *PIF3* expression in the embryo. a** Protein gel blot analysis of PIF3 levels in WT embryos and endosperms. Seeds were cultivated under white light (50 μmol/m²/s) at 22 and 34 °C and dissected at the indicated times upon seed imbibition. PIF3 antibody control: protein extract from WT and *pif3* seedlings grown for 2 days in darkness. Numbers represent PIF3 levels normalized to those of UGPase. **b** Representative pictures of transgenic embryos and endosperms carrying a *PIF3* promoter fused to GUS dissected at the indicated times upon seed imbibition and stained with X-Gluc. Seeds were cultivated under white light (50 μmol/m²/s) at 22 °C or 34 °C in absence (−ABA) or presence (+ABA) of 3 μM ABA. Bar: 0.1 mm for all pictures except MS 48 h (bar: 0.25 mm), 72 h (bar 0.5 mm) and 96 h (bar 0.75 mm). **c** PIF3 protein levels at the indicated times in WT embryos dissected 4 h after seed imbibition and cultivated under white light (50 μmol/m²/s) at 22 °C or 32 °C in absence (MS) or presence of 3 μM ABA. Numbers represent PIF3 levels normalized to those of UGPase. **d** Protein gel blot analysis of PIF3 levels in WT and *phyB* embryos that were either dissected 4 h after seeds imbibition and cultivated for 3 days (−) or dissected after seeds were cultivated for 3 days (+) at 22 °C or 28 °C. Numbers represent PIF3 levels normalized to those of UGPase. Errors bars represent standard deviations.

Under high temperatures, we found that 48 h upon seed imbibition, *pif3* endosperm has lower expression in genes associated with RNA metabolism and ribosome biogenesis whereas it has higher expression in genes associated with cell wall modifications (Supplementary Data 1). These observations might be consistent with the notion that *pif3* endospermic cells are in a distinct metabolic state relative to that of WT endospermic cells. Indeed, in *pif3* seeds, which lack thermoinhibition, the *pif3* endosperm does not repress germination and is therefore bound to degenerate as it is abandoned by the seedling after germination. This could explain the drop in RNA metabolism gene expression. On the other hand, the increase in cell wall modification gene expression might reflect that the *pif3* endosperm cells are bound to detach from each other to allow the embryonic radicle to emerge from the seed (germination).

Seed thermoinhibition can only be observed in non-dormant seeds since dormant seeds will not germinate under favorable conditions, i.e. imbibition at 22 °C in presence of white light (normal conditions). In the Col-0 accession, dormancy is lost after a couple of weeks of dry after-ripening, i.e. two-week old WT Col-0 seeds will germinate at 22 °C. Interestingly, seed thermoinhibition is strongest in younger WT seed batches. For example, a two-week-old WT seed batch will be fully thermoinhibited at 28 °C, unlike a two-month-old WT seed batch. If thermoinhibition results from higher temperatures lowering phyB signaling in the endosperm and if thermoinhibition at a given temperature becomes weaker with older seeds, then the model predicts that phyB-mediated germination ought to gain strength as seed age. This notion is indeed supported by experiments performed by De Giorgi et al. using WT seeds of various seed ages[36]. Indeed, in two-week-old seeds able to germinate under normal conditions, a pulse of red light early upon seed imbibition followed by incubation in darkness, which normally leads to phyB-mediated germination, is insufficient to promote germination[36,37]. In two-month-old seeds, the same treatment only partially triggers germination and full germination is only observed in 6-month-old WT seeds[36].

If thermoinhibition is the result of lower phyB signaling, then one would expect *phyB* mutant seeds to never germinate even under normal conditions. This is indeed the case to some degree: two-week old *phyB* mutants are unable to germinate under normal conditions (22 °C), unlike WT seeds. Full *phyB* germination under normal conditions is only observed in two-month-old *phyB* seeds. In this study, we therefore systematically used seed batches that are at least two-month-old so that WT and *phyB* germination is comparable under normal germination conditions. Two-month-old *phyB* seeds could germinate under normal conditions but not under darkness, as previously reported (Supplementary Fig. 6a)[37,38]. This suggests that the other phytochromes promote seed germination at 22 °C under normal conditions and high temperatures would also accelerate their reversion to their inactive state in *phyB* mutants exposed to 28 °C thus leading to *phyB* mutant thermoinhibition[12,13,39]. Accordingly, two-month-old *phyA phyB* (*phyAB*) and *phyB phyC phyD phyE* (*phyBCDE*) seeds did not germinate at 22 °C (Supplementary Fig. 6b) whereas three-year-old *phyAB* seeds germinated like WT seeds and *phyBCDE* seeds did not germinate well (Supplementary Fig. 6c). Furthermore, three-year-old *phyAB* and *phyBCDE* seeds were thermoinhibited at 28 °C (Supplementary Fig. 6c). However, the contribution of the other

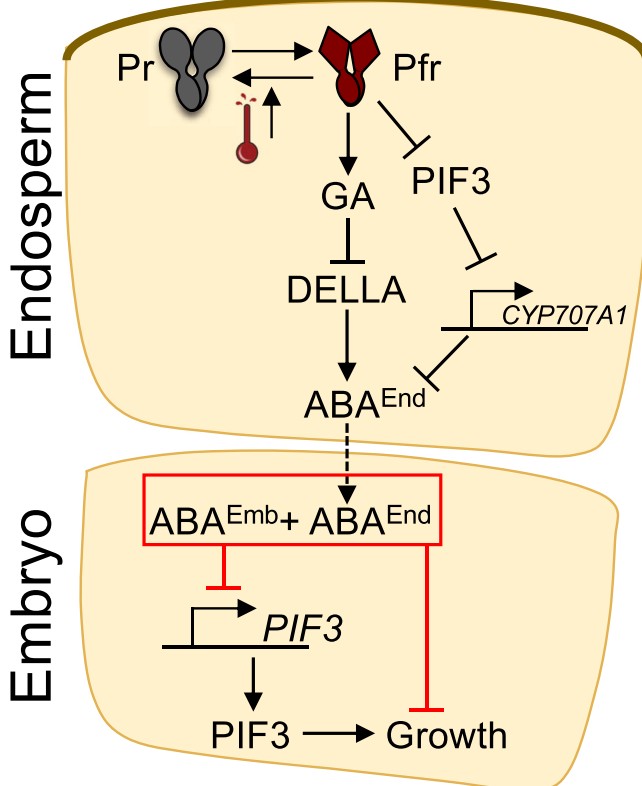

**Fig. 6 | A model for seed thermoinhibition in *Arabidopsis thaliana*.** High temperatures accelerate thermal reversion of phyB, which leads to lower levels of its Pfr signaling active form. This activates two parallel pathways promoting endospermic ABA accumulation (ABA^End) and release towards the embryo to promote, together with embryonic ABA (ABA^Emb), thermoinhibition by repressing growth, which includes repressing the expression of embryonic *PIF3* whose product promotes growth. In one pathway, GA synthesis is repressed, a process that may involve PIF factors (not shown in the model), which leads to stabilization of DELLA factors that promote ABA accumulation. In another pathway, stabilization of endospermic PIFs leads to increased endospermic ABA accumulation. The model shows the case of PIF3, which promotes endospermic ABA accumulation and release by repressing *CYP707A1* expression.

phytochromes for seed thermoinhibition could only be revealed in absence of *phyB* as *phyACDE* seeds behaved similar to WT seeds (Supplementary Fig. 6b, c). Altogether, these observations indicate that temperature-driven phyB inactivation plays an essential role to promote thermoinhibition, which is reinforced by the inactivation of the remaining phytochromes.

Previous work provided indirect evidence that endospermic PIF1 promotes ABA accumulation and release from the endosperm to repress germination in response to far-red light[14]. Here, we showed that PIF3 indeed promotes ABA accumulation and release under higher temperatures to promote thermoinhibition. Hence, the work presented here helps understanding a paradox concerning how PIFs regulate early seedling development. Indeed, whereas PIFs promote

growth in seedlings, some PIFs, such as PIF1 and PIF3, repress embryonic growth in seeds. This paradox is partly resolved if one considers that the role of PIFs to repress germination reflects their activity to promote ABA accumulation and release in the endosperm rather than their activity to promote growth in the embryo. Interestingly, PIF3-mediated endospermic ABA release maintains embryonic *PIF3* expression low thus inhibiting embryonic PIF3-mediated growth. Hence, low endospermic phyB signaling hijacks low embryonic phyB signaling. Our work here consolidates the importance of phyB signaling in the endosperm to control the onset of the embryo-to-seedling transition and further reveals the importance of PIFs as essential phyB signaling components also in the endosperm. phyB signaling was mainly studied and characterized in seedlings where it regulates numerous developmental processes[10,11]. The study of endospermic phyB signaling and the identification of its specificities will enable to understand how evolution has tinkered with phyB-mediated signaling pathways to implement germination control mechanisms in angiosperms, producing endosperm-bearing seeds, while retaining adaptive phyB signaling developmental responses in seedlings.

## Methods

### Plant material

The Arabidopsis mutant and transgenic seeds used in this study (Col-0 background) were described previously: *aba1-6*[40], *della* (*rgl2-SK54 rga-28 gai-t6 rgl1-SK62 rgl3-3*)[41], *phyA-211*[42], *phyB-9*[43], *phyB-9 phyC-2 phyD-201 phyE-201* and *phyA-211 phyC-2 phyD-201 phyE-201*[44], *pif3-1* and *pif3-3*[30], *pif3-3 phyB-9*, *pif1-1 pif3-3* and *pif1-1 pif3-3 pif5-3*[45], *pif1-1*[15], *pif4-2* and *pif7-1*[45], *pif5-3*[46] and *pif6-1* (SALK_090239C). *phyB-9/35S:phyB-YFP* and *phyB-9 /35S:phyB*$^{S86D}$*-YFP*[21], WT/*35S:phyB-GFP* and WT/*35S:phyB*$^{G564E}$*-GFP*[22], WT/*promPIF3:GUS*[32]. The *phyB della* and *pif3 della* mutants were created using CRISPR/Cas9 editing of the *della* mutant. The T3 generation was analyzed after crossing out the Cas9 transgene in the T2 generation. The *PHYB* mutant allele has an A insertion at position 603 leading to a frameshift and the *PIF3* mutant allele has C insertion at position 1847 leading to a frameshift. Western blot analysis confirmed the absence of phyB and PIF3 protein in *phyB della* and *pif3 della* mutants, respectively (Supplementary Fig. 7)

### Plant growth conditions and germination assays

The *Arabidopsis* mutant and transgenic seeds were harvested on the same day from plants grown under the same conditions and afterripened at room temperature for the same time period. Seeds were surface sterilized by agitation for 10 min in a 1.5 % solution of sodium hypochlorite containing 0.05% Tween followed by three washing steps using sterile water.

For the germination assays under white light, seeds were surface sterilized and sown on plates containing MS medium (Sigma) and 0.8% (w/v) agar and incubated for 1 h at 22 °C under white light (0.7 μmol/m²/s). Thereafter, the plates were incubated under white light (50 μmol/m²/s) at the indicated temperatures.

For the germination assays after FR or R light pulse irradiation, seeds were surface sterilized and incubated for 2 h under white light (0.7 μmol/m²/s). Thereafter, seeds were irradiated in a dark chamber by a 5 min pulse of FR light (3.69 μmol m⁻² s⁻¹) was followed or not by 5 min pulse of R light (14.92 μmol m⁻² s⁻¹). The irradiated seeds were retrieved from the dark chamber located in a dark room equipped with safe green light and wrapped under three layers of aluminium foil. The wrapped plates were incubated at the indicated temperatures.

### Dissection and growth conditions of dissected material

Seeds were surface sterilized and incubated for 4 h at 22 °C under white light (0.7 μmol/m²/s). Thereafter, embryos and endosperms were dissected in MS medium (Sigma) containing 0.8% (w/v) agar and the SCBA was assembled as previously described under a binocular emitting white light (500 μmol/m²/s)[47,48]. The maximum time exposure to

binocular white light of the seed material undergoing dissection, including the dissected endosperms used to measure endospermic ABA levels and release, was 2 h. SCBAs used 30 endosperms for 1 embryo or 100 endosperms for 4 embryos (Fig. 2c, d). Plates containing the dissected embryos or SCBAs were incubated under white light (50 μmol/m²/s) at the indicated temperatures. For endospermic ABA levels and release, dissected endosperms were resuspended in 300 μl of water and irradiated with FR or FR/R light as described for seeds. Dissected embryo material used for protein analysis (e.g. as in Fig. 1e without the End+ material) was frozen in liquid nitrogen at the indicated times prior to perform RNA or protein extraction procedures.

Concerning seed embryo or endosperm material used for RNA and protein analysis (e.g. Fig. 3b) seeds were surface sterilized and incubated for 1 h at 22 °C under white light (0.7 μmol/m²/s). Then, the plates were incubated under white light (50 μmol/m²/s) at the indicated temperatures. At the indicated time of analysis, embryos and endosperms were dissected under a binocular emitting white light (500 μmol/m²/s) for no longer than 30 min at 22 °C and then frozen in liquid nitrogen. Hypocotyl lengths were measured at the indicated times using Fiji[49].

### Generation of *phyB della* and *pif3 della* using CRISPR/Cas9

For the generation of CRISPR/Cas9 mutant alleles of *PHYB* and *PIF3* in the *della* mutant, the method described by Tsutsui and Higashiyama was used[50]. For targeting the 5′ end of *PHYB*, the primers 5′- attgatctgttgctcaggtacag −3′ and 5′- aaacctgtacctgagcaacagat −3′ were annealed and cloned into the pKIR1.1 plasmid. For targeting the 5′ end of *PIF3*, the primers 5′-attgagcaatatccatcaaggga −3′ and 5′-aaactcccttgatggatattgct-3′ were used.

### RNA extraction and RT-qPCR

Total RNA was extracted from 200 endosperms or embryos as previously described[51]. Total RNA was treated with RQ1 RNase-Free DNase (Promega) and reverse-transcribed using ImProm II reverse transcriptase (Promega) and oligo(dT)15 primer. Quantitative RT-PCR was performed using Quant Studio 5 real-time PCR system (Applied Biosystems) and GoTaq qPCR Master Mix (Promega). Relative transcript level was calculated using the comparative Δ Ct method and normalized to the *PP2A* gene transcript levels. The primers to amplify *CYP707A1* (At4g19230) are 5′-tcatctcaccaccaagta-3′ and 5′-aaggcaattctgtcattcta −3′[52]; *PIF3* (At1g09530) 5′-gacggcgtgataggatcaac-3′ and 5′-catcgaagctttgtccacct-3′; *PIF4* (At2g43010) 5′-aagtcgaaccaacgatcagg-3′ and 5′-ttgcaaagccttcattctctc-3′[53]; *PP2A* (At1g69960) 5′-ggaccggagccaactagga-3′ and 5′-gctatccgaacttctgcctcatt-3′[14].

### Histochemical GUS staining

Histochemical GUS staining assay was performed using a substrate buffer: 100 mM sodium phosphate (pH 7.0), 5 mM potassium ferricyanide, 5 mM potassium ferrocyanide, 1 mM EDTA, 1% Triton-X, 1 mg/mL X-Gluc for embryo and 5 mg/mL for endosperm samples. Samples were incubated at 37 °C for 16 h and washed after the staining 3 times with 75% ethanol.

### Antibody production and protein gel blot

PIF3 recombinant proteins (full-length) were prepared using PIF3-his DNA (pET28C, Novagen) provided by Ferenc Nagy and induced and purified using commercial kit (HisTrap Fast Flow, GE17-5319-01). Polyclonal anti-PIF3 was obtained from rabbits immunized with PIF3 recombinant protein (Cocalico Biologicals Inc., USA). PIF3 antibodies were further affinity-purified using PIF3 recombinant protein immobilized on nitrocellulose filters as described[6]. Endosperms and embryos were homogenized in presence of protein extraction buffer (50 mM Tris-HCl pH 6.8, 2% SDS, 10% glycerol, 100 mM DTT, 0.01% bromophenol blue) and protein extract corresponding to 150

endosperms or 15 embryos was loaded per lane. Proteins were separated on 10% SDS-PAGE gel and transferred to a PVDF membrane (Amersham). Membranes were incubated in 5% milk powder in Tris-buffered saline (TBS) containing PIF3 antibody (1:200 dilution), RGL2 antibody (1:250 dilution)[6] or ABI5 antibody (1:2000 dilution)[19] for 16 h at 4 °C. Anti- UGPase antibody (Agrisera) was used at 1:10,000 dilution and anti-PHYB antibody and used at 1:1,000 dilution for 16 hours at 4 °C. Anti-rabbit (for PIF3, RGL2, ABI5 and UGPase) or anti-mouse (for PHYB) IgG HRP-linked whole antibody (GE healthcare) in a 1:10,000 dilution was used as secondary antibody for 2 h at RT. Membranes were washed with TBS + 0.05% Tween 3 times for 10 min after the first and second antibody incubation and the immune complexes were detected using the ECL kit (Amersham). All the protein accumulation data shown in main figures and supplementary figures arise from the same blots. Protein signals were quantified digitally using the ImageJ software (v 2.0) and using UGPase signal as a normalization factor. For each blot all values are relative to the strongest signal found in the blot which is arbitrarily set to 1. Western blot repetitions can be found in Supplementary Fig. 8. Where portions of blots have been presented in the main figures, the full blots are shown in the Supplementary Fig. 9.

### RNA-seq and gene expression profile analysis

Total RNA was isolated from 200 endosperms dissected 40 h after seeds incubation at 30 °C and from 200 embryos dissected 4 h after seeds imbibition and growing for 40 h at 30 °C. Total RNA was isolated as previously described[51]. cDNA libraries were prepared from 200 ng of total RNA using a TruSeq mRNA Library Prep Kit (Illumina). cDNA libraries were normalized and pooled then sequenced using HiSeq 2500 (Illumina) with single-end 50 bp reads. Transcript assembly and normalization was performed with the Cufflinks program and gene expression levels were calculated in FPKM (Fragments Per Kilobase of exon per Milion mapped fragments) units. Differential gene expression analysis was performed by Cuffdiff, a part of the Cufflinks package[54–56].

GO enrichment analysis for biological processes was performed using the TAIR publicly available tool (https://www.arabidopsis.org/tools/go_term_enrichment.jsp). The differentially expressed genes (DEGs) with log$_2$ fold change ≥1 or ≤1and $P$ ≤ 0.05 were selected for GO enrichment analysis. The list of genes bound by PIF3 was previously described Zhang et al.[32]. Gene lists for abscisic acid metabolic process, abscisic acid-activated signaling pathway, gibberellin metabolic process, and gibberellin mediated signaling pathway were obtained from TAIR. Heat maps were generated using the heatmap.2 function of the gplots package in R.

### Sample preparation protocol for ABA measurement

For the extraction of supernatants, 10 ul of ABA-d6 (100 ng/mL in water) were added to the collected supernatant and the mixture was lyophilized overnight. The dry residue was resuspended in 100 ul of methanol 35%., vortexed and sonicated for 1 min each, and the suspension was transferred in a 0.2 mL PCR tube and centrifuged at 14'000 g for 2 min. The supernatant was collected and placed in an HPLC vial fitted with a conical insert for analysis. The extraction of plant tissues was done as follows. The tissue was lyophilized overnight in a 2.0 mL Eppendorf tube. Then the tube was frozen in liquid nitrogen and ground into powder using two 3 mm stainless steel beads in a mixer mill (30 Hz, 15 s). 990 ul of ethylacetate:formic acid (99.5:0.5, v/v) and 10 ul of ABA-d6 (100 ng/mL in water) were added and the tube was shaken for 3 min at 30 Hz. After centrifugation at 14'000 g for 3 min, the supernatant was transferred to a new 2.0 mL tube, and the pellet from the first tube was re-extracted using 0.5 mL of ethylacetate:formic acid (99.5:0.5, v/v). Both supernatants were combined and evaporated to dryness at 35 °C in a centrifugal evaporator. The residue was resuspended in 100 ul of methanol 35%, vortexed and sonicated for 1 min each, and the suspension was transferred in a 0.2 mL PCR tube and centrifuged at 14'000 g for 2 min. The supernatant was collected and placed in an HPLC vial fitted with a conical insert for analysis.

### ABA measurement

ABA measurements were performed using a protocol adapted from Glauser et al.[57].

The analysis of ABA was done by HPLC-MS/MS using a QTRAP 6500+(Sciex) connected to an Acquity UPLC (Waters). 3.7 ul of extract was injected onto an Acquity UPLC BEH C18 column (50 × 2.1 mm, 1.7 um particle size, Waters). A gradient of phase A - H$_2$O:formic acid 0.05% and phase B - acetonitrile:formic acid 0.05% from 5–65% B in 6.5 min was applied. The flow rate was set to 0.4 mL/min and the column temperature to 35 °C. The mass spectrometer was operated in electrospray negative ionization using the multiple reaction monitoring (MRM) mode. Transitions for ABA and ABA-d6 were 263/153 and 269/159, respectively. For both ABA and ABA-d6, declustering potential (DP), collision energy (CE) and collision cell exit potential (CXP) were set to −45 V, −14 V and −17 V, respectively. The ion spray voltage was set to −4500 V, the source gas temperature to 500 °C, and GS1, GS2 and curtain gas flows to 40, 35 and 35 psi, respectively. A five-point calibration curve (0.02, 0.1, 0.5, 5 and 20 ng/mL, all containing ABA-d6 at 10 ng/mL) was used for quantification. Linear regressions weighted by 1/x were applied. Analyst v.1.7.1 was used to control the instrument and for data processing. We ran blank samples and 0.02 ng/mL ABA standards over the run of the batches as quality controls.

Data availability statement: the authors declare that the data supporting the findings of this study are available within the article, its Supplementary Information and data.

### Reporting summary

Further information on research design is available in the Nature Portfolio Reporting Summary linked to this article.

## Data availability

Source data are provided with this paper. The gene expression data have been published in Gene Expression Omnibus under accession number GSE224926. LC-MS/MS data used for ABA measurements are available upon request. Source data are provided with this paper.

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

## Acknowledgements

We thank Pablo Cerdan for providing *phyACDE*, *phyBCDE* and *phyAB* seeds, Elena Monte for providing *pif3* mutants and a transgenic line carrying a *PIF3* promoter fused to GUS seeds, Ferenc Nagy for providing the PIF3-HIS DNA plasmid, *phyB* mutant seeds complemented with PHYB that has a WT or Ser86Asp or Gly564Glu sequences, Akira Nagatani for providing phyB antibody and Ted Farmer for providing *aos* and *coi1-34* seeds. We would like to thank Mylene Docquier and members of the Genomics Platform of the Institute of Genetics and Genomics (iGE3) at the University of Geneva for help with RNAseq experiments. This work was supported by grants from the Swiss National Science Foundation (No 152660) and by the State of Geneva.

## Author contributions

U.P, M.S., M.I, G.G, L.L.-M. designed experiments. U.P, M.S., M.I, G.G, L.L.-M. performed the experiments and analyzed the data. U.P and L.L.-M. wrote the paper. All authors reviewed and approved the manuscript.

## Competing interests

The authors declare no competing interests.
