## [Peer Review File · Nature Communications]

The Arabidopsis endosperm is a temperature-sensing tissue that implements seed thermoinhibition through phyBReviewer #1 (Remarks to the Author):

In this manuscript, the author first used a seed coat bedding assay (SCBA) to test the effect of endosperm on the embryo germination under high temperature, and find endosperm is important to sense phyB signal, HT inactivates phyB to stabilize PIF3, which repress the ABA degradation gene CYP707A1 to promoter endosperm ABA biosynthesis, the accumulated ABA release into embryo to block embryo germination, meanwhile ABA repress the embryonic PIF3 accumulation, which should promoter embryonic growth. At last, the authors propose that PIF3 play the opposite function in embryo and endosperm. After all the manuscript show interesting to suggest the separate function of embryo and endosperm, and suggest the opposite role of PIF3 in embryo and endosperm, however, we have several concerns that should be fully addressed. However, we worry about PIF3 is the main and sole factor to control seed germination inhibition under HT, and how to accurately define the function of endosperm PIF3 and embryonic PIF3, because the author underline the opposite function of PIF3 in endosperm and embryo, how to explain such opposite function of PIF3 in different tissue.

Main concerns:

1) the SCBA method: the author: the authors peel the whole seed and separate the embryo from endosperm, such progress should damage the seed vigor and induce some specific wound-related phytohormone generation, such as jasmonic acid, which also affect the seed germination, thus the author should design some experiment to evaluate the unexpected effect on seed germination, such as jasmonic-acid related mutant such as *coi1*, *jaz* mutant et al.

2) the function of PIF1 during HT. Their previous study (Keun Pyo Lee et al, *Gene Dev*,2012) reveal the important of PIF1, rather than PIF3, in control seed germination in light-initiated seed germination and HT stress, though the author also used the *pif1* mutant (supplemental figure 2) to exclude the role of PIF1 in HT-mediated seed thermoinhibition, but they used the whole seeds but not the separated embryo to test the effect HT on *pif1* mutant. What is more, we compared the different seed germination among different *pif* mutant in Figure 4A, in batch1 and batch2 experiment, we still found the high seed germination of *pif1* and *pif5* than wild type Col line, and in all batch experiments, *pif3/4* mutant display higher seed germination than *pif3* single mutant, suggesting the redundant function of *pif4* during germination thermoinhibition. In supplement figure 2, the seed germination of *pif1/3*, *pif3/4* and *pif1/3/5* was higher than *pif1* single mutant under 32C treatment, suggesting the function of *pif1* and *pif5* in regulating seed germination thermoinhibition.

3)Figure 1f: as for the dissect endosperm, but not measured the ABA content in the separated embryo, thus cannot obtain the result that embryo still synthesized ABA during HT stress, though they used *aba2-1* mutant, but it is not enough.

4) The spatial expression of PIF3, the author provide the PIF3:GUS evidence to show that HT induce the endosperm PIF3 expression at transcriptional level, but phyB mainly affect the protein stability of PIF3 rather than PIF3 transcripts, thus detecting PIF3 protein expression in endosperm is better. As we mentioned above, how to explain its opposite function in endosperm and embryo.

5)the author did not present molecular or genetic evidence to define the function of CYP707A1 to support their model. I think CYP707A2, rather than CYP707A1 is more important to determine seed germination, thus why the author select CYP707A1, but not CYP707A2.

Minor concern

1) as *phyb* mutant present longer hypocotyl under normal or HT condition, thus hypocotyl length is a good index to reflect the seed germination status under HT. Seed germination percentage is enough to directly reflect the seed germination.

2) Beside *phyB*, *phyA* also play the function in controlling seed germination, how about *phyA* in controlling seed germination under HT, though the author show provide evidence to show that *phyA* is not necessary for seed germination thermoinhibition under HT.

Reviewer #2 (Remarks to the Author):

Seeds are comparably well protected against adverse environmental conditions while seedlings are very delicate and vulnerable. Plants therefore tightly control the germination of seeds so that germination is completed only under conditions optimal for seedling establishment. Temperature is a critical environmental factor that controls seed germination and seedling growth. Elevated temperatures inhibit the germination of seeds in different species. This phenomenon, referred to as thermoinhibition, is of considerable relevance for ecology and agriculture. In the manuscript, the authors investigate the molecular mechanism underlying thermoinhibition in the model species *Arabidopsis thaliana*. Using a set of very elegant experiments, they show that thermoinhibition is not autonomously controlled by the embryo but is strongly dependent on the endosperm, a tissue surrounding the embryo. They identify a critical role of ABA, DELLAs, and PIF3 in this process. A role of ABA is possibly not unexpected but – at least to my knowledge – PIF3, in contrast to PIF1, has not been implicated in control of germination so far. Interestingly, high temperatures lead to stabilisation of PIF3 in the endosperm, triggering the accumulation of ABA which then results in reduced levels of PIF3 in the embryo. Furthermore, they provide evidence that thermal reversion of endospermic phyB plays a role in sensing the temperature. Overall, this is a very exciting manuscript presenting novel data that help understand how plants control the regulation of germination to avoid germination under temperature conditions that are not favourable for subsequent seedling establishment.

Comments:

- 1) The model (Figure 6) suggests that only the endosperm is relevant for thermoinhibition while the embryo itself does not contribute. In my view, however, this is not consistent with data shown in Figure 1b. The genotype of the embryo clearly affects hypocotyl length at 34 °C. WT on WT seedlings are much shorter than WT on *aba1* and *aba1* on WT seedlings, showing that a defect in ABA biosynthesis in the endosperm but also in the embryo impairs thermoinhibition. Thus, it is true that the endosperm (and ABA derived from the endosperm) is required for thermoinhibition, but it is equally true that ABA synthesised in the embryo is required (this is also mentioned at the end of the respective paragraph in the text). Please adjust the model in Figure 6 so that it is consistent with data in Figure 1.
- 2) The reference to light conditions used in the study that is given in the method section is not very useful (i.e. it does not really explain the light treatments). However, knowing the exact light conditions is absolutely critical to interpret the data presented in the manuscript. Therefore, please do not refer to other publications and instead describe for each experiment the exact light treatment, including light treatment/conditions during imbibition, light treatment/conditions during/after dissection, light treatment/conditions during incubation between dissection and evaluation of hypocotyl growth, ...
- 3) Data presented in the manuscript are in line with a role of phyB as thermosensor for thermoinhibition of germination but I wonder if the authors also considered other potential thermosensors. ELF3 – for instance – is temperature-sensitive. Is anything known about a potential role of ELF3 in regulation of germination?
- 4) Using phyB as thermosensor only works if seeds are exposed to light to convert phyB to Pfr, which then reverts to Pr at a rate that depends on the temperature. However, several species – also species closely related to *Arabidopsis* such as mustard – germinate independently of light. I wonder how widespread thermoinhibition of germination is and if it correlates with light-requirement for germination? How could species that show thermoinhibition but germinate independently of light sense the temperature (if such species exist)? Any comments, ideas or suggestions?

Minor comments:

- 1) Introduction: "... In the Pr state PIFs are stabilized, which promotes their accumulation...", "... the signaling-inactive Pr state promotes the stabilization of phytochrome-interacting factors (PIFs) that promote growth ...", "... Inactive phyB promotes ABA synthesis ...", and similar statements implicating that the inactive state of phyB has a function should be revised. PIFs are stabilised if

phyB is in the Pr state but PIFs are equally stabilised if phyB is not present at all (such as in a phyB mutant); same is true for other responses. Thus, the critical point is that Pfr is absent and not that Pr is present.

2) Introduction: "In the endosperm, high temperature reduces the pool of active phyB, which activates endospermic ABA synthesis and release.": This would implicate that primarily ABA synthesis is regulated but in the following sentence, the authors say that expression of an ABA catabolic gene is downregulated. Can the authors comment on that/clarify this?

3) Is pifq already an established name for the pif13457 mutant? If not, why not use pifp (pif pentuple mutant) for pif13457 instead of pifq (pif quintuple) so that it is different from the established name (pifq) for the pif1345 quadruple mutant?

4) Please indicate the light conditions (not only the temperature) in all figure legends.

5) Discussion: "... and high temperatures would also accelerate their reversion to their inactive state in phyB mutants exposed to 28 °C thus leading to phyB mutant thermoinhibition.". Please give a reference for this statement.

6) Figure 1b: I) n = 15 is indicated in the figure legend but only 12 data points are shown in the chart; please clarify. II) What does "t0" mean?

7) Figure 1f: I) Please provide additional details regarding statistical analysis. What was compared to what? Did you correct for multiple comparison?

8) Figure 2b: I) n = 12 is indicated in the figure legend but more than 12 data points are shown for phyB.

9) Figure 2c: I) n = 10 is indicated in the figure legend but number of data points is not 10 for some samples.

10) Figure 2e: Please provide additional details regarding statistical analysis. What was compared to what? Did you correct for multiple comparison? $p < ?$

11) Figure 2e, f: Why use 30 °C and 32 °C (instead of 34 °C) in Figure 2e and f? Is the phenotype less pronounced at 34 °C than at 30/32 °C? ... fine for me; I just wonder.

12) Figure 3d: I) Please define far-red light pulse and red light pulse (fluence or time and fluence rate). II) Please provide additional details regarding statistical analysis. What was compared to what? Did you correct for multiple comparison? $p < ?$ III) Three seed batches are mentioned in the figure legend but only two data points (or even only one) are shown for some samples.

13) Figure 3e: I) Did you dissect the embryos in safe green light? II) Three independent experiments are mentioned in the figure legend but only two data points are shown for FR/della. III) Please provide additional details regarding statistical analysis. What was compared to what? Did you correct for multiple comparison? $p < ?$

14) Figure 3f: Is the unit for hypocotyl length correct, i.e. is it cm and not mm?

15) Figure 4c: Please indicate the light conditions.

16) Figure 4e: I) n = 3 is indicated in the figure legend but 2 or >3 data points are shown for some samples. II) Please provide additional details regarding statistical analysis. What was compared to what? Did you correct for multiple comparison? $p < ?$

17) Figure 5a, b, c: Please indicate the light conditions.

18) Methods: Are there no washing steps in the histochemical GUS staining?

19) Methods: I) Did you use full-length PIF3 for immunisation? II) "... purified using commercial kit (Amersham).": what kit was used? III) "Membranes were incubated in 5% milk in Tris-buffered saline (TBS) containing PIF3 antibody (5:1,000 dilution), RGL2 antibody (4:1,000 dilution) or ABI5 antibody (5:10,000 dilution)": 5% milk powder or really 5% milk? pH of TBS? 5:1000, 4:1000, 5:10000 dilution means 1:200, 1:250, and 1:2000, respectively, correct?

20) Supplementary Figure 1a: Please provide additional details regarding statistical analysis. What was compared to what? Did you correct for multiple comparison? $p < ?$

21) Supplementary Figure 1c: I) Three independent batches are indicated in the figure legend but 4 data points are shown for samples. II) Please provide additional details regarding statistical analysis. What was compared to what? Did you correct for multiple comparison? $p < ?$

22) Supplementary Figure 1e: Three independent batches are indicated in the figure legend but 4 data points for samples.

23) Supplementary Figure 2a, b: Please indicate the light conditions.

24) Supplementary Figure 2b: 3-4 independent batches are indicated in the figure legend but only one or two data points are shown for some samples (32 °C/pif5, 32 °C+GA/pif1 and pif1/3).

25) Supplementary Figure 2c: n>10 embryos is indicated in the figure legend but only 8 data points are shown for 28 °C/pif7.

26) Supplementary Figure 4: Does the figure show data from a single biological replicate with

technical replicates? Please indicate this clearly or otherwise indicate the number of biological replicates.

27) Supplementary Figure 5: Why do Col-0 seeds germinate in the dark? Weren't they treated with a FR pulse after sowing? Please indicate the light treatments.

Reviewer #3 (Remarks to the Author):

Authors report that highT inhibits seed germination by inhibiting phyB in the endosperm. The inhibition of endosperm phyB causes increases in ABA level in the endosperm, which is then pumped out and inhibits embryo growth. Authors demonstrate this by performing SCBA assay. The results are very interesting. I have a few comments.

1. SCBA assay is very interesting assay, but I'm curious if this is specific to the endosperm. Have authors tested any other macerated tissues instead of ruptured endosperm to prove the effect is specific to endosperm? Though authors have published a few papers with this assay, I don't think authors have shown that the effect is specific to endosperm.

2. Authors are showing ABI5 protein levels in various figures to show ABA released from the endosperm (due to highT or phyB mutation) increases ABI5 protein level in embryo. However, this could be associated with the different developmental stages, i.e. all embryos having high ABI5 level are not developing from 'seed' stage embryo, whereas all embryos having the lower ABI5 level are actively growing. Authors are measuring ABI5 protein levels late enough to observe the different growth between two comparing embryos. I think authors may need to measure ABI5 protein level earlier.

3. Authors need to measure RGL2 protein level earlier to decouple it from the change in developmental stages.

4. Similarly, authors need to prove that the expression of PIF3 at lowT but not at highT is not associated with the developmental change.

5. RNA-seq was performed at 48 hours after the imbibition. Authors wrote there is no observable difference between WT and pif3 at this time point. However, if we magnify seeds in the supplemental figure 3, we can see that the pif3 mutant seeds already undergo testa rupturing whereas WT seeds do not. Thus, the huge difference in RNA-seq data of the pif3 mutant also could be due to the developmental stage difference.

6. It is difficult to follow author's logic of claiming that PIF3 is the most important PIF mediating highT for seed germination. Germination phenotype of pif1, pif3 and pif5 single mutant is very similar (Fig 4a, Supple Fig 2). Authors also showed similar germination phenotypes of pif1/pif3, pif3/pif4 and pif1/pif3/pif5 multiple mutants. To claim that PIF3 is the major PIF, author need to show that pif1/5 and pif1/4 mutants have lower germination frequencies at highT than pif3/5 and pif3/4 mutants.

7. [Minor] pifp mutant is written as pifq mutant.

8. [minor] please show comparing immunoblot in the same blot (e.x. Fig 5a-endosperm, PIF3 at 22 and 34, Fig 5d) or quantify.

Reviewer #4 (Remarks to the Author):

In this manuscript, Piskurewicz and coworkers studied the role of the endosperm in thermoinhibition, or inhibition of seed germination at elevated temperature. They monitored germination of embryos with or without the endosperm (seed coat bedding assay), and used various mutants affected in phytochrome and hormone signaling. They demonstrated that thermoinhibition is mediated by inactivation of PhyB in the endosperm, which promotes accumulation of endospermic PIF3. PIF3 increases endospermic ABA synthesis, which leads to

inhibition of embryo growth. In the embryo, elevated ABA inhibit PIF3, which would otherwise promote embryo growth.

The authors have used seed coat bedding assays in the past to dissect the role of the endosperm in seed germination. In this study, they have uncovered a prominent role for the endosperm in promoting thermoinhibition through phytochrome and ABA/GA signaling. The study is well done, and the results are interesting, however there is a large overlap with previous findings.

The positive and negative roles of ABA and GA synthesis/signaling in thermoinhibition, as well as the role of far-red light in inhibition of germination by elevated temperature, have been previously shown. Furthermore, the role of endospermic ABA in seed germination inhibition has been previously shown, although at control temperature. Previous work also showed that the endosperm inhibits germination under FR light and the same authors used a seed coat bedding assays to dissect the role of Phy and ABA signaling in the embryo and endosperm, although at control temperature. Phytochromes were also shown to play a role in seed germination across various temperatures. Last, the role of elevated temperature in promoting Phy inactivation was shown during hypocotyl growth. Overall, these findings decrease the novelty of this study. REFS (also included in this papers): Hershchel et al 2007; Toh et al., 2008; Piskurewicz et al 2009; Lee et al 2010, 2012; etc.

Other comments:

1) Although RNAseq shows a minimal overlap in genes regulated by PIF3 in embryo and endosperm, suggesting different roles of PIF3 in these tissues, no follow-up of the comparative transcriptomic analysis is done aside from showing gene ontology enrichment. Therefore, it is unclear what these results mean without molecular, genetic, and functional validation.

2) Changes (or lack of) in protein levels need to be normalized and quantified in replicated experiments. Ideally, when comparing protein levels in different mutant backgrounds or under different growth conditions (temperature), protein extracts should be loaded on the same blot. All blots shown are cropped and it is difficult to compare protein levels. Examples: Fig 2d; 3b,c,f and 5a,c,d.

3) phyB/della and pif3/della mutants generated by CRISPR/Cas9: which generation (T2/T3) of mutants was characterized? was Cas9 was crossed out? Were PHYB transcripts analyzed in mutants?

REVIEWER COMMENTS

We would like to thank the reviewers for their effort in evaluating our work and for their constructive comments and criticisms. We have carefully read and addressed all the comments made by the reviewer in the revised manuscript. In blue you will find the original comments made by the reviewers while our response is in black.

Reviewer #1 (Remarks to the Author):

In this manuscript, the author first used a seed coat bedding assay (SCBA) to test the effect of endosperm on the embryo germination under high temperature, and find endosperm is important to sense phyB signal, HT inactivates phyB to stabilize PIF3, which repress the ABA degradation gene CYP707A1 to promoter endosperm ABA biosynthesis, the accumulated ABA release into embryo to block embryo gemination, meanwhile ABA repress the embryonic PIF3 accumulation, which should promoter embryonic growth. At last, the authors propose that PIF3 play the opposite function in embryo and endosperm. After all the manuscript show interesting to suggest the separate function of embryo and endosperm, and suggest the opposite role of PIF3 in embryo and endosperm, however, we have several comments that should be fully addressed. However, we worry about PIF3 is the main and sole factor to control seed germination inhibition under HT, and how to accurate define the function of endosperm PIF3 and embryonic PIF3, because the author underline the opposite function of PIF3 in endosperm and embryo, how to explain such opposite function of PIF3 in different tissue.

These issues are addressed below when we answer to your major concerns.

Main concerns:

1) the SCBA method: the author: the authors peel the whole seed and separate the embryo from endosperm, such progress should damage the seed vigor and induce some specific wound-related phytohormone generation, such as jasmonic acid, which also affect the seed germination, thus the author should design some experiment to evaluate the unexpected effect on seed germination, such as jasmonic-acid related mutant such as *coi1*, *jaz* mutant et al.

You raise a legitimate concern. It seems, however, that wound-related germination effects are not significant enough to explain our observations. Indeed, we make the following observations: 1) Under high temperatures, the WT seed does not germinate (thermo-inhibition), 2) removing the endosperm triggers embryonic growth and 3) culturing the embryos on a bed of endosperms (SCBA) prevents embryonic growth. One could argue that endosperm removal (observation 2) triggers embryonic growth due to the stress of the dissection procedure. However, in the context of a SCBA, the endosperm and embryonic tissues were also obtained after a dissection procedure and yet the embryo does not grow under high temperatures. This weakens the argument that the observed embryonic growth in absence of the endosperm is a result of stress. One could argue that, when dissected, the endosperm is stressed and releases a germination repressive activity that is unrelated to thermo-inhibition. This seems unlikely because under normal temperatures the dissected endosperms will not prevent the growth of embryos (see Fig. A below).

As per your suggestion, we tested the behavior of *jaz* mutant seeds. We found that the *jaz* seed, endosperm and embryo behave like WT seed material under normal or high temperatures (Figure A). This further supports the conclusion that the endosperm exerts an

activity that represses the growth of the embryo under high temperatures unrelated to wounding signaling.

Hypocotyl length was measured three days after imbibition at 34°C. "WT End+" is the hypocotyl length of the embryo within the thermoinhibited seed. "WT End-" is the length of the embryo cultured without endosperm (dissection 4h after seed imbibition). For the SCBAs, the WT dissected embryo were cultured on a bed of WT or *jaz1* endosperms. Please note that the seed batch used was at least 1 year old, which explains with the SCBA using WT endosperm was less efficient to repress growth in comparison to the endosperms used throughout the manuscript which came from younger seed batches. Lower case letters above histograms are used to establish whether two values are statistically significantly different as assessed by one-way ANOVA followed by a Tukey HSD test ($p < 0.05$): different letters denote statistically different values.

2) the function of PIF1 during HT. Their previous study (Keun Pyo Lee et al, Gene Dev, 2012) reveal the important of PIF1, rather than PIF3, in control seed germination in light-initiated seed germination and HT stress, though the author also used the *pif1* mutant (supplemental figure 2) to exclude the role of PIF1 in HT-mediated seed thermoinhibition, but they used the whole seeds but not the separated embryo to test the effect HT on *pif1* mutant. What is more, we compared the different seed germination among different *pif* mutant in Figure 4A, in batch1 and batch2 experiment, we still found the high seed germination of *pif1* and *pif5* than wild type Col line, and in all batch experiments, *pif3/4* mutant display higher seed germination than *pif3* single mutant, suggesting the redundant function of *pif4* during germination thermoinhibition. In supplement figure 2, the seed germination of *pif1/3*, *pif3/4* and *pif1/3/5* was higher than *pif1* single mutant under 32C treatment, suggesting the function of *pif1* and *pif5* in regulating seed germination thermoinhibition.

In Lee et al. we only studied PIF1 in the context of FR-dependent germination arrest (we did not address the role of PIF1 under high temperatures). In this manuscript, we are examining the role of various PIFs and we did not intend to exclude that PIF1 plays a role in seed thermoinhibition. In fact, we show that it does together with other PIFs. The data in Figure 4A and suppl fig 2c are meant to show that several PIFs can promote thermoinhibition, particularly PIF1, PIF3 and PIF5. We also clearly stated that numerous PIFs are contributing (e.g. in the results we wrote "Altogether, these data show that PIFs, and notably PIF3, promote seed thermoinhibition.").

To avoid conveying that PIF3 is the only PIF promoting thermoinhibition, we have modified the text of the title, the abstract, some parts of the results and the legend of the model to avoid potential misunderstandings (e.g. we now write "Altogether, these data show that PIFs, and notably PIF1, PIF3 and PIF5, promote seed thermoinhibition."). The reason we think PIF3 is interesting and deserves a more detailed characterization is because it was 1)

not previously involved in temperature responses, 2) not previously involved in the control of seed germination (unlike PIF1) and 3) it promotes antagonistic early embryonic growth responses in the endosperm and the embryo, which is not the case for other PIFs, particularly at high temperatures (Figure 4b). These arguments are now explicitly stated in the results section to justify our focus on studying PIF3 function during thermoinhibition.

3)Figure 1f: as for the dissect endosperm, but not measured the ABA content in the separated embryo, thus cannot obtain the result that embryo still synthesized ABA during HT stress, though they used *aba2-1* mutant, but it is not enough.

Please note that our work focuses on understanding the endospermic signaling pathway controlling endospermic ABA synthesis rather than the embryonic pathway. This is motivated by the presented evidence that thermoinhibition is crucially dependent on the endosperm. Nevertheless, we thought it would be interesting to show our available data indicating that the embryo also synthesizes ABA under high temperatures to promote thermoinhibition. It is challenging to measure the real contribution of embryonic ABA synthesis during thermoinhibition. Indeed, if one dissects embryos at a given time point upon imbibition to measure their endogenous ABA levels, one will not be able to discriminate the ABA contributed by the endosperm from the ABA contributed by the embryo. To further provide evidence for the claim that the embryo indeed increases ABA synthesis under high temperatures, we measured embryonic *NCED9*, *NCED2* and *CYP707A1* expression 36h after imbibition, a time where germination has not yet taken place. *NCEDs* are ABA biosynthesis genes whereas *CYP707A1* is an ABA catabolic gene. We found that *NCEDs* and *CYP707A1* expression markedly increase and decrease, respectively, under high temperatures relative to normal temperatures (Fig. B). Although indirect, these data provide further evidence that the embryo increases its endogenous ABA synthesis during thermoinhibition relative to non-thermoinhibited embryos. We think these additional data are interesting and are now included in the manuscript (supplementary Figure 1a) (In response to reviewer 2, we also included the contribution of embryonic ABA synthesis into the final model in Fig. 6). However, since these data do not affect the main conclusions of our work, the data concerning embryonic ABA synthesis could also in principle be removed.

Figure B

Relative *NCED2*, *NCED9* and *CYP707A1* mRNA accumulation in WT embryos dissected from WT seeds after 36h upon seed imbibition under white light (50 $\mu\text{mol}/\text{m}^2/\text{s}$) at indicated temperatures. Total RNA was isolated from 100 embryos. Histograms show the average expression value from 3 technical replicates. Expression levels were normalized to those of *PP2A*.

4) The spatial expression of PIF3, the author provide the PIF3:GUS evidence to show that HT induce the endosperm PIF3 expression at transcriptional level, but *phyB* mainly affect the protein stability of PIF3 rather than PIF3 transcripts, thus detecting PIF3 protein expression in endosperm is better. As we mentioned above, how to explain its opposite function in endosperm and embryo.

We did measure PIF3 protein levels in the endosperm, the results were shown in Fig. 5A. They show that endospermic PIF3 protein levels increase at high temperatures relative to low temperatures (Fig. 5A). We compared endospermic PIF3 levels in WT and *phyB* mutants (Figure C). As expected, at high temperatures (32°C), PIF3 levels are higher in the *phyB* mutant endosperm relative to WT (please note that it is difficult to detect PIF3 protein in the WT endosperm, in this experiment performed at 32°C, the signal was barely detectable) (Figure C).

PIF3 protein levels in WT and *phyB* endosperms dissected from seeds at the indicated times upon seed imbibition. Seeds were cultured at 32°C under white light (50 $\mu\text{mol}/\text{m}^2/\text{s}$). Numbers represent PIF3 levels normalized to those of UGPase.

Our data show that embryonic PIF3 protein highly accumulates when the embryos are transforming themselves into seedlings at 22°C. *pif3* shows shorter hypocotyls in germinated seedlings indicating that embryonic PIF3 promotes early seedling growth. At 34°C, embryonic PIF3 protein levels remain low over time. Furthermore, we showed that ABA prevents the embryonic PIF3 accumulation observed at 22°C, most likely by repressing *PIF3* mRNA accumulation (Fig. 5b). These results suggest that high temperatures block PIF3 transient accumulation by keeping the early embryonic state. By contrast in the endosperm, PIF3 protein was barely detectable at 22°C but its levels increased at 34°C, likely as the combined result of *PIF3* mRNA induction and PIF3 protein stabilization. Since *phyB* regulates thermoinhibition, mainly in the endosperm (Fig. 2), these observations are consistent with the model that high temperatures increase PIF3 protein by inhibiting *phyB* signaling in the endosperm.

In summary, high temperatures induce endospermic PIF3 accumulation to repress germination; however, PIF3 promotes early seedling growth in germinating seeds at 22°C or at high temperatures that are not sufficient to completely block germination in a given seed population. In other words, endospermic PIF3 regulates a distinct developmental process relative to embryonic PIF3 which translates into distinct (opposite) growth behavior in the seedling.

Concerning PIF3 opposite function in endosperm and embryo, we performed RNA-seq and showed that PIF3 regulates the expression of different gene sets. This is not surprising given that the endosperm tissue is distinct from the embryo so that endospermic PIF3 function is

modulated by the distinct molecular environment distinguishing the endosperm cells from the embryonic cells (different hormonal activities, different transcription factors etc..).

5)the author did not present molecular or genetic evidence to define the function of CYP707A1 to support their model. I think CYP707A2, rather than CYP707A1 is more important to determine seed germination, thus why the author select CYP707A1, but not CYP707A2.

The reason we selected *CYP707A1* is because endospermic *CYP707A1* expression is strongly upregulated in *pif3* mutants under high temperatures unlike that of *CYP707A2*, which is not significantly affected. Hence, PIF3 regulates *CYP707A1* expression and not that of *CYP707A2*.

We respectfully disagree about considering *CYP707A2* more important than *CYP707A1* regarding their role in the control of seed germination. Previous work by the lab of Nambara clearly shows that mutating *cyp707a1* or *cyp707a2* leads to marked increases in ABA levels in seeds (Fig1b of Okamoto et al 2006) and marked delays in seed germination (Fig. 1c of Okamoto et al. 2006). It is true that *cyp707a1* mutants germinates slightly better than *cyp707a2* mutants (a difference of about 10% of the seed population over the course of 8 days upon imbibition (Fig1C of Okamoto et al 2006), but clearly both genes have a strong impact on seed germination as indeed concluded by the group of Nambara.

We have genetic evidence that *cyp707a1* is important for thermoinhibition and genetic evidence shows that *cyp707a1* is epistatic to *pif3*, suggesting that *CYP707A1* functions downstream of PIF3 (see our response to comment #1 by reviewer #4).

Minor concern

1) as *phyb* mutant present longer hypocotyl under normal or HT condition, thus hypocotyl length is a good index to reflect the seed germination status under HT. Seed germination percentage is enough to directly reflect the seed germination.

When we score germination we score radical protrusion out of the seed coat, which is the definition of germination *sensu stricto* (e.g. Fig. 4C of Weitbrecht et al, Journal of Experimental Botany, Vol 62, issue 10, June 2011, Pages 3289-3309). However, in the context of the SCBA it is difficult to assess protrusion out of the seed coat since there is no seed coat. Hence, we measure hypocotyl length to monitor germination. The pictures that accompany the hypocotyl measurements are representative of what we observe.

2) Beside *phyB*, *phyA* also play the function in controlling seed germination, how about *phyA* in controlling seed germination under HT, though the author show provide evidence to show that *phyA* is not necessary for seed germination thermoinhibition under HT.

The role of *phyA* and of the other phytochromes is discussed in the discussion, which refers also to data present in Supplementary Fig. 4b and 4c. We show (see suppl. fig 4c) that *phyA* can play a role to regulate seed thermoinhibition and that this *phyA* contribution can be observed in a *phyB* mutant background. However, our work focuses on *phyB* given that *phyB* mutant has a clear seed thermoinhibition phenotype.

Reviewer #2 (Remarks to the Author):

Seeds are comparably well protected against adverse environmental conditions while seedlings are very delicate and vulnerable. Plants therefore tightly control the germination of seeds so that germination is completed only under conditions optimal for seedling establishment. Temperature is a critical environmental factor that controls seed germination and seedling growth. Elevated temperatures inhibit the germination of seeds in different species. This phenomenon, referred to as thermoinhibition, is of considerable relevance for ecology and agriculture. In the manuscript, the authors investigate the molecular mechanism underlying thermoinhibition in the model species *Arabidopsis thaliana*. Using a set of very elegant experiments, they show that thermoinhibition is not autonomously controlled by the embryo but is strongly dependent on the endosperm, a tissue surrounding the embryo. They identify a critical role of ABA, DELLAs, and PIF3 in this process. A role of ABA is possibly not unexpected but – at least to my knowledge – PIF3, in contrast to PIF1, has not been implicated in control of germination so far. Interestingly, high temperatures lead to stabilisation of PIF3 in the endosperm, triggering the accumulation of ABA which then results in reduced levels of PIF3 in the embryo. Furthermore, they provide evidence that thermal reversion of endospermic phyB plays a role in sensing the temperature. Overall, this is a very exciting manuscript presenting novel data that help understand how plants control the regulation of germination to avoid germination under temperature conditions that are not favourable for subsequent seedling establishment.

We are glad you found this work interesting.

Comments:

1) The model (Figure 6) suggests that only the endosperm is relevant for thermoinhibition while the embryo itself does not contribute. In my view, however, this is not consistent with data shown in Figure 1b. The genotype of the embryo clearly affects hypocotyl length at 34 °C. WTonWT seedlings are much shorter than WTonaba1 and aba1onWT seedlings, showing that a defect in ABA biosynthesis in the endosperm but also in the embryo impairs thermoinhibition. Thus, it is true that the endosperm (and ABA derived from the endosperm) is required for thermoinhibition, but it is equally true that ABA synthesised in the embryo is required (this is also mentioned at the end of the respective paragraph in the text). Please adjust the model in Figure 6 so that it is consistent with data in Figure 1.

As per your suggestion, we have modified the model to include the contribution of embryonic ABA synthesis. Regarding the contribution of the embryo, please see also our response to comment #3 by Reviewer 1 (we have performed embryonic ABA biosynthesis and catabolic gene expression measurements further indicating that ABA synthesized in the embryo is required for thermoinhibition).

2) The reference to light conditions used in the study that is given in the method section is not very useful (i.e. it does not really explain the light treatments). However, knowing the exact light conditions is absolutely critical to interpret the data presented in the manuscript. Therefore, please do not refer to other publications and instead describe for each experiment the exact light treatment, including light treatment/conditions during imbibition, light treatment/conditions during/after dissection, light treatment/conditions during incubation between dissection and evaluation of hypocotyl growth, ...

Thank you for pointing this out. We have updated the materials and methods section providing a detailed explanation of the manipulations performed and the light treatments at every step (prior to dissection -if applicable-, during dissection and after dissection-if applicable-). These procedures are always the same throughout this work and their detailed description occupies substantial amount of text space. Therefore, we think it is not advisable to include all the light conditions in all the figure legends (your minor point 4). We have nevertheless indicated in the figure legends the light conditions during the incubation period of the seed material (it is always 50 $\mu\text{mol}/\text{m}^2/\text{s}$) or the R and FR light intensities.

3) Data presented in the manuscript are in line with a role of phyB as thermosensor for thermoinhibition of germination but I wonder if the authors also considered other potential thermosensors. ELF3 – for instance – is temperature-sensitive. Is anything known about a potential role of ELF3 in regulation of germination?

We focused our work on phyB and we cannot exclude that there are other thermosensing mechanisms in seeds besides phyB and the other phytochromes (see discussion and supplementary Fig. 4b and 4c concerning the role of the other phytochromes). As per your suggestion, we evaluated seed thermoinhibition responses in *elf3* mutant seeds (Figure D). We used 3 independent WT seed batches and 6 independent *elf3-1* (Col) seed batches (n=50 seeds were tested for each batch) and germination was scored after 3 days at 22°C, 28°C and 32°C. We found that *elf3* seeds were moderately, but significantly thermoinhibited at 28°C, unlike WT seeds (* p<0.05). Thus, ELF3 could play a thermosensing role in seeds although our data suggest that this role is weaker relative to that of phyB.

Figure D

4) Using phyB as thermosensor only works if seeds are exposed to light to convert phyB to Pfr, which then reverts to Pr at a rate that depends on the temperature. However, several species – also species closely related to Arabidopsis such as mustard – germinate independently of light. I wonder how widespread thermoinhibition of germination is and if it correlates with light-requirement for germination? How could species that show thermoinhibition but germinate independently of light sense the temperature (if such species exist)? Any comments, ideas or suggestions?

You raise very interesting questions.

Concerning your first question, the short answer is that thermoinhibition and light-requirement for germination among species have not been sufficiently investigated to be

able to tell whether thermoinhibition and light-requirement for germination are correlated or not. According to MacDonald and Hart (MacDonald, I.R. and J.W. Hart, *Ann Bot* **47**, 275-277) who cite Black (Black, 1969, Light-controlled germination of seeds. In *Dormancy and Survival*, ed. H. W. Woolhouse, pp. 193-217, Symp. Soc. exp. Biol. no. 23, Cambridge University Press, London.): “Although the germination of many seeds seems to be indifferent to the presence of light, the majority of species show responses varying from inhibition to stimulation or even an absolute light requirement (Black, 1969). However, it can be misleading to categorize seed behaviour in such a way, since the light responses of many species have been shown to be markedly influenced by other factors in the environment (Vidaver, 1977).” (Vidaver, W., 1977. Light and seed germination. In *The Physiology and Biochemistry of Seed Dormancy and Germination*, ed. A. A. Khan, pp. 181-192, Elsevier/North-Holland Biomedical Press). We agree with the latter sentence: we have never worked with mustard seeds, but we know from our experience in *Arabidopsis* that besides temperature seed germination is influenced by other environmental factors such as water potential, nutrient availability (e.g. phosphate or sucrose can promote germination) or the duration of treatments that release dormancy (such as the duration of seed stratification). In addition, seed germination responses are further influenced by the age of the seed. Hence, different pathways influence germination and environmental factors with light (and temperature) being one of them. Depending on those factors, light may exert different germination responses in a given species (e.g. a pulse of red light followed by darkness will not promote germination in young *Arabidopsis* seeds but will promote germination in older seeds). The decision to germinate or not is of paramount importance for the plant so it is expected that in each species evolution has tinkered with the various pathways controlling seed germination. So, it is possible that in a given species, seeds may seemingly be not affected by light when tested under conditions that happen to be optimal to promote the other pathways promoting germination. Hence, each case must be studied carefully. Evidence for these claims can be found in the literature. For example, in the case of mustard that you mention, I.R. MacDonald and J.W. Hart (same reference as above) noticed that white light could inhibit seed germination under a low water potential, which is not in itself inhibitory to germination in the dark: “...we have observed that a low water potential, which is not in itself inhibitory to germination in the dark, becomes inhibitory in the [white] light”. So, it would seem that mustard seed germination can be repressed by light under certain conditions. We checked the literature for thermoinhibition of mustard seeds under low water potential, but we could not find a paper explicitly dealing with this. That light may exert opposite germination responses in two Brassicaceae is consistent with the notion that control of seed germination is under strong evolutionary pressure. If the observations by MacDonald and Hart are confirmed, mustard would not be an isolated case: the group of Mittelsten-Scheid has shown that germination in *Aethionema arabicum*, another Brassicaceae, is inhibited by light and this is mediated by phytochromes (bioRxiv, <https://doi.org/10.1101/2022.06.24.497527>). Mérai et al (Mérai et al *Journal of Experimental Botany*, 2019, Vol. 70, No12 pp 3313-3329) showed that, when active, *Aethionema arabicum* phytochromes promote ABA signaling and repress GA signaling. Hence, evolution has kept the sensing mechanism but tinkered with the signaling pathways controlling germination downstream of phytochromes. Mérai et al also showed that light-dependent repression of *Aethionema arabicum* germination requires, as in *Arabidopsis*, the presence of the endosperm. It will be interesting to investigate thermoinhibition in *Aethionema arabicum*.

Concerning your second question, let us assume the interesting case where we identified a species for which we cannot find conditions to observe light-dependent germination responses and let us assume that this species displays seed thermoinhibition. In this case, one could imagine a phytochrome that has evolved to no longer sense light but instead has evolved to be in equilibrium between an active and inactive signaling conformation with the ratio of active-to-inactive pool being set by temperature.

Minor comments:

1) Introduction: "... In the Pr state PIFs are stabilized, which promotes their accumulation...", "... the signaling-inactive Pr state promotes the stabilization of phytochrome-interacting factors (PIFs) that promote growth ...", "... Inactive phyB promotes ABA synthesis ...", and similar statements implicating that the inactive state of phyB has a function should be revised. PIFs are stabilised if phyB is in the Pr state but PIFs are equally stabilised if phyB is not present at all (such as in a phyB mutant); same is true for other responses. Thus, the critical point is that Pfr is absent and not that Pr is present.

Thank you for bringing this up. We have modified the text to make it clear that is it the absence or the reduction of the Pfr pool that stabilizes PIFs rather than the presence of Pr.

2) Introduction: "In the endosperm, high temperature reduces the pool of active phyB, which activates endospermic ABA synthesis and release.": This would implicate that primarily ABA synthesis is regulated but in the following sentence, the authors say that expression of an ABA catabolic gene is downregulated. Can the authors comment on that/clarify this?

Thank you for pointing this out. Our phrasing lacks precision. Endogenous ABA levels are the result of the rates of ABA synthesis and catabolism. An increase in ABA levels can reflect a decrease in catabolism and not necessarily an increase in synthesis. So, in the revised manuscript we wrote "In the endosperm, high temperature reduces the pool of active phyB, which increases endospermic ABA levels and release". In the same vein, we changed the wording throughout the manuscript (e.g. "Our results indicate that low endospermic phyB signaling promotes endospermic ABA accumulation and release..." instead of "Our results indicate that low endospermic phyB signaling promotes endospermic ABA synthesis and release.." (Third sentence in the discussion).

3) Is *pifq* already an established name for the *pif13457* mutant? If not, why not use *pifp* (*pif* pentuple mutant) for *pif13457* instead of *pifq* (*pif* quintuple) so that it is different from the established name (*pifq*) for the *pif1345* quadruple mutant?

Thank you for pointing this out. *pifq* is the established name for *pif1345*. The quintuple mutant (*pif1 pif3 pif4 pif5 pif7*) was first described by Zhang et al. 2020 (PNAS, 117(6) 3261-3269) and named *pifq pif7* and further abbreviated as *pifq7*. So, we propose to keep this nomenclature (*pifq7*) in this manuscript (we have modified the text and figure accordingly).

4) Please indicate the light conditions (not only the temperature) in all figure legends.

See our comment to your point 2 above. We have indicated in the figure legends the white light intensity during the incubation period of the seed material or the R and FR light intensities.

5) Discussion: "... and high temperatures would also accelerate their reversion to their inactive state in phyB mutants exposed to 28°C thus leading to phyB mutant thermoinhibition.". Please give a reference for this statement.

We added the two references that provide the first evidence that phytochromes function as thermosensors (using phyB)(Jung et al. 2016 and Legris et al. 2016).

6) Figure 1b: I) $n = 15$ is indicated in the figure legend but only 12 data points are shown in the chart; please clarify. II) What does “ t_0 ” mean?

I) This is indeed a mistake, the figure legend is corrected. ii) t_0 is the time of dissection of the WT seed (i.e. 4h after seed imbibition); this is now specified in the figure legend.

7) Figure 1f: I) Please provide additional details regarding statistical analysis. What was compared to what? Did you correct for multiple comparison?

We redid the statistical analysis for this figure. In the results section we wrote “..endosperms cultured at 30°C for 48h contained 8-fold more ABA than those cultured at 22°C (Fig. 1f). Furthermore, over the same period, the levels of ABA in the culture medium at 30°C were more than 16x higher than those at 22°C (Fig. 1f).” Hence, we used one-way ANOVA followed by a Tukey HSD test ($p < 0.05$) to assess on one hand whether ABA levels in the endosperm (End) 40h after incubation at 22°C and 30°C are statistically different (they are, $p < 0.05$) and on the other hand whether ABA levels in the culture medium (Sup) at 22°C and 30°C are statistically different (they are, $p < 0.05$). We did not seek to assess whether ABA levels in the supernatant and endosperm are statistically different. We have modified the figure and the figure legend accordingly.

8) Figure 2b: I) $n = 12$ is indicated in the figure legend but more than 12 data points are shown for phyB.

Yes, sorry about that. We corrected the figure legend ($n=12$ for WT embryos and $n=16$ for *phyb* embryos).

9) Figure 2c: I) $n = 10$ is indicated in the figure legend but number of data points is not 10 for some samples.

Sorry about this, n ranges from 7 to 10. The figure legend is corrected.

10) Figure 2e: Please provide additional details regarding statistical analysis. What was compared to what? Did you correct for multiple comparison? $p < ?$

For the hypocotyl length we used one-way ANOVA followed by a Tukey HSD test ($p < 0.05$). For the germination percentage we redid the statistical analysis using one-way ANOVA followed by a Tukey HSD test ($p < 0.05$)(multiple comparison). The figure legend is corrected accordingly. We also redid the statistical analysis for the germination percentage in Figure 3f as in Figure 3e.

11) Figure 2e, f: Why use 30 °C and 32 °C (instead of 34 °C) in Figure 2e and f? Is the phenotype less pronounced at 34 °C than at 30/32 °C? ... fine for me; I just wonder.

In Figure 2e and 2f we are using temperatures that allow to assess the occurrence of hypersensitive and hyposensitive thermoinhibition responses, respectively. Hence, in Figure 2e, if we had used 34°C then the phyB control would be thermoinhibited, which would prevent us to detect higher thermoinhibition in phyB^{S86D} since all phyB and phyB^{S86D} seeds would be arrested. In Figure 2f, the temperature of 32°C was selected as it reveals the thermoinhibition hyposensitivity of the phyB^{S86E} line in the same number of days as in Figure 2e.

12) Figure 3d: I) Please define far-red light pulse and red light pulse (fluence or time and fluence rate). II) Please provide additional details regarding statistical analysis. What was compared to what? Did you correct for multiple comparison? $p < ?$ III) Three seed batches are mentioned in the figure legend but only two data points (or even only one) are shown for some samples.

I) The figure legend now contains the information regarding the FR and R pulses.
II) We performed the statistical analysis again using one-way ANOVA followed by a Tukey HSD test ($p < 0.05$) (multiple comparison). The Figure legend is corrected accordingly.
III) Sorry for the confusion, the software we use (Prism 8) may only display one dot (or one shadowy dot) depending on the width of the column. It is only for the conditions of 30°C R and FR that there are two seed batches. The rest of the conditions have three seed batches. Thus, we manually added the dots to display the missing values. The figure is now corrected.

13) Figure 3e: I) Did you dissect the embryos in safe green light? II) Three independent experiments are mentioned in the figure legend but only two data points are shown for FR/della. III) Please provide additional details regarding statistical analysis. What was compared to what? Did you correct for multiple comparison? $p < ?$

I) Figure 3e shows ABA released by dissected endosperms. They were not dissected under safe green light. Endosperms are dissected under white light and then suspended in water. Thereafter they are irradiated with FR or FR/R pulses. The manipulation procedure and the light conditions at every step are now clearly specified in the materials and methods. The figure legend provides more information regarding the procedure and contains the information about light intensities.

II) Thank you for picking this up, we indeed omitted to say that the FR *della* condition has two biological replicates. This is now corrected in the figure legend.

III) We have performed the statistical analysis again. We used one-way ANOVA followed by a Tukey HSD test to assess on one hand whether ABA levels released by the WT and *della* endosperm treated by R light are statistically different (they are, $p < 0.001$) and on the other hand whether ABA levels released by the WT and *della* endosperm treated by FR light are statistically different (they are, $p < 0.05$). We did not seek to compare values obtained with R and FR light treatments. We have modified the figure and the figure legend accordingly.

14) Figure 3f: Is the unit for hypocotyl length correct, i.e. is it cm and not mm?

It indeed should be mm, the figure is now corrected.

5) Figure 4c: Please indicate the light conditions.

The light conditions are specified in the figure legends. We also performed the statistical analysis again using one-way ANOVA followed by a Tukey HSD test.

16) Figure 4e: I) $n = 3$ is indicated in the figure legend but 2 or >3 data points are shown for some samples. II) Please provide additional details regarding statistical analysis. What was compared to what? Did you correct for multiple comparison? $p < ?$

I) Same issue as above, we have manually added the dots for the missing values. The condition 32°C R and FR *pif3/della* has 4 biological replicates and the rest has three biological replicates. The Figure legend is corrected accordingly.

II) We performed the statistical analysis again using one-way ANOVA followed by a Tukey HSD test ($p < 0.05$) (multiple comparison). The Figure legend is corrected accordingly.

17) Figure 5a, b, c: Please indicate the light conditions.

The light conditions are now indicated in the figure legend.

18) Methods: Are there no washing steps in the histochemical GUS staining?

Yes, indeed there are; the materials and methods now specify the washing steps.

19) Methods: I) Did you use full-length PIF3 for immunisation? II) "... purified using commercial kit (Amersham).": what kit was used? III) "Membranes were incubated in 5% milk in Tris-buffered saline (TBS) containing PIF3 antibody (5:1,000 dilution), RGL2 antibody (4:1,000 dilution) or ABI5 antibody (5:10,000 dilution)": 5% milk powder or really 5% milk?

pH of TBS? 5:1000, 4:1000, 5:10000 dilution means 1:200, 1:250, and 1:2000, respectively, correct?

I) Yes, we used full-length PIF3.

II) We specified which kit we used in the materials and methods.

III) Yes, you are correct about the dilutions, this is now how the dilution values are now specified in the material and methods section. We used indeed 5% milk powder, thank you for picking up this, this is now specified in the material and methods section.

20) Supplementary Figure 1a: Please provide additional details regarding statistical analysis. What was compared to what? Did you correct for multiple comparison? $p < ?$

This is now Supplementary Figure 1b. Please note that again the program we use masks the dots so we added them manually. We performed the statistical analysis again using one-way ANOVA followed by a Tukey HSD test ($p < 0.05$) (multiple comparison). The Figure legend is corrected accordingly.

21) Supplementary Figure 1c: I) Three independent batches are indicated in the figure legend but 4 data points are shown for samples. II) Please provide additional details regarding statistical analysis. What was compared to what? Did you correct for multiple comparison? $p < ?$

I) This is now Supplementary Figure 1d. Indeed, 4 biological samples are shown, we have corrected the figure legend

II) We performed the statistical analysis again using one-way ANOVA followed by a Tukey HSD test ($p < 0.05$) (multiple comparison). The Figure legend is corrected accordingly.

22) Supplementary Figure 1e: Three independent batches are indicated in the figure legend but 4 data points for samples.

I) This is now Supplementary Figure 2a. Indeed, 4 biological samples are shown, we have corrected the figure legend. We performed the statistical analysis again using one-way ANOVA followed by a Tukey HSD test ($p < 0.05$) (multiple comparison). The Figure legend is corrected accordingly.

23) Supplementary Figure 2a, b: Please indicate the light conditions.

This is now Supplementary Figure 2b. The light conditions are now indicated in the figure legend.

24) Supplementary Figure 2b: 3-4 independent batches are indicated in the figure legend but only one or two data points are shown for some samples (32 °C/pif5, 32 °C+GA/pif1 and pif1/3).

Sorry again. This is now Supplementary Figure 2c. In absence of exogenous GA: For *pif 1* and *pif5* there are indeed two biological replicates (we manually added one dot for *pif5* 32°C and 2 dots for *pif1/pif3* at 32°C+GA). In presence of exogenous GA: for *pif1/3* there are two biological replicates.

25) Supplementary Figure 2c: $n > 10$ embryos is indicated in the figure legend but only 8 data points are shown for 28 °C/pif7.

This is now Supplementary Figure 2d. The figure legend is corrected.

26) Supplementary Figure 4: Does the figure show data from a single biological replicate with technical replicates? Please indicate this clearly or otherwise indicate the number of biological replicates.

Thank you. This is indeed a single biological replicate with technical replicates. This figure has been removed and no longer commented in the discussion because we do not want to emphasize only the expression of *PIF3* relative to that of *PIF4*.

27) Supplementary Figure 5: Why do Col-0 seeds germinate in the dark? Weren't they treated with a FR pulse after sowing? Please indicate the light treatments.

This is now Supplementary Figure 4. This experiment was done for the purpose of the discussion where we discuss the role of other phytochromes. So, we did not treat seeds with a FR pulse, which would inactivate other phytochromes -except phyA who plays a role to promote germination at later time points upon imbibition- in order to suggest that phytochromes other than phyB promote germination at 22°C under white light. Indeed, if old enough (e.g. 2 months) Col-0 seeds will germinate in the dark unless treated with a FR pulse. Germination in darkness is promoted by phyB (Shinomura et al. 1994 Plant Physiol. 104(2):363-371; Shinomura et al. 1996 PNAS 93 (15) 8129-8133). These references are now mentioned in the discussion.

Reviewer #3 (Remarks to the Author):

Authors report that highT inhibits seed germination by inhibiting phyB in the endosperm. The inhibition of endosperm phyB causes increases in ABA level in the endosperm, which is then pumped out and inhibits embryo growth. Authors demonstrate this by performing SCBA assay. The results are very interesting. I have a few comments.

1. SCBA assay is very interesting assay, but I'm curious if this is specific to the endosperm. Have authors tested any other macerated tissues instead of ruptured endosperm to prove the effect is specific to endosperm? Though authors have published a few papers with this assay, I don't think authors have shown that the effect is specific to endosperm.

One cannot exclude that other plant vegetative tissues could influence embryonic growth. However, unlike the endosperm, those tissues (e.g. a leaf) are not normally closely associated with embryos so if they would have an effect on embryonic growth one would have to think about the interpretation of such observation. We observe that when the endosperm is removed, the embryos are no longer thermoinhibited, suggesting that the endosperm is necessary for thermoinhibition (see also our comments to reviewer 1 regarding other interpretations). In our SCBA experiments we observe embryonic growth responses that are consistent with those of the undissected seed: WT endosperms arrest WT embryos at high temperatures but not at low temperatures. This allows us to reasonably assume that the SCBA can be used to dissect the genetic pathways operating in the endosperm to implement seed thermoinhibition. As per your suggestion, we examined other tissues and we found that they did not have a statistically significant effect on embryonic growth under normal or high temperatures (Figure E).

WT embryos dissected 4h upon seed imbibition were cultured on endosperms, petals, roots and cotyledons at the indicated temperatures for three days when hypocotyl lengths were measured (more than 10 embryos were used in each condition). Roots were isolated from 2 week old seedlings. Lower case letters above histograms are used to establish whether two values are statistically significantly different as assessed by one-way ANOVA followed by a Tukey HSD test ($p < 0.05$): different letters denote statistically different values.

2. Authors are showing ABI5 protein levels in various figures to show ABA released from the endosperm (due to highT or phyB mutation) increases ABI5 protein level in embryo. However, this could be associated with the different developmental stages, i.e. all embryos having high ABI5 level are not developing from 'seed' stage embryo, whereas all embryos having the lower ABI5 level are actively growing. Authors are measuring ABI5 protein levels late enough to observe the different growth between two comparing embryos. I think authors may need to measure ABI5 protein level earlier.

ABI5 is used as a developmental marker for germination arrest.

Embryonic growth arrest is the developmental output of the signaling pathway promoting thermoinhibition (we describe the growth arrest by providing a representative picture together with hypocotyl length measurements). It is known that ABA promotes ABI5 accumulation in the embryo and if sufficient ABA levels are present then ABA will block embryonic growth over time upon imbibition, which is associated with sustained ABI5 accumulation (e.g. Fig.1B Piskurewicz et al. 2008). Hence, ABI5 accumulation is a well-established marker for an ABA-mediated growth arrest. Therefore, high ABI5 accumulation in thermoinhibited seeds provides strong support to the notion that the growth arrest observed in thermoinhibited seeds is due to ABA accumulation.

So, for example, when we observe in Fig. 1d that a WT embryo cultured on a bed of WT endosperms remains arrested 3-4d after imbibition whereas the same WT embryo is not arrested when cultured on a bed of *aba1* endosperms, we conclude that the embryonic arrest is due to the endospermic ABA release that blocks embryonic growth. The western

blot is just meant to further support this conclusion by showing that indeed the arrested embryo has higher ABI5 protein relative to the non-arrested embryo. Hence, the ABI5 western is a molecular confirmation that that growth arrest was indeed triggered by ABA. It is not useful to examine ABI5 levels at earlier time points (e.g. 12h, 24h) because ABI5 protein levels are high irrespective of whether the embryo will continue growing or not [Indeed, ABI5 is highly abundant in dry seeds, see for example Fig. 1B, left panel labeled "Normal" or Fig. 7B left panel of Piskurewicz et al 2008, Vol. 20: 2729-2745)]. Whether the embryo will be arrested or not depends on whether it will sustain or not high ABA levels over time upon imbibition.

3. Authors need to measure RGL2 protein level earlier to decouple it from the change in developmental stages.

RGL2 is used as a marker for germination arrest involving low GA conditions.

RGL2 is a repressor of seed germination under low GA conditions as *rgl2* mutants germinate under low GA conditions unlike WT seeds (Lee et al 2002 Genes Dev 16(5):646-658).

Sustained RGL2 accumulation over time in arrested seeds is a marker of seeds arrested as a result of low GA levels (see Fig. 1D condition "PAC" of Piskurewicz et al 2008). In turn, RGL2 promotes ABA accumulation which promotes and sustains the germination arrest over time (see Fig. 7 of Piskurewicz et al 2008). Since phyB signaling promotes GA synthesis in seeds and since the hypothesis states that phyB signaling is repressed in response to high temperatures, we asked whether thermoinhibited seeds behave as seeds having low GA levels. In this context, RGL2 overaccumulation in thermoinhibited seeds is used as a marker to support the notion that the thermoinhibition pathway indeed lowers GA levels and that DELLAs are promoting thermoinhibition by promoting ABA accumulation. The latter is supported by the observation that *della* mutants are less thermoinhibited than WT (e.g. suppl Fig1b) and that *della* endosperms release less ABA (Fig. 3a and 3e).

Concerning RGL2 protein accumulation, under normal conditions that lead to germination due to increased GA synthesis upon imbibition, RGL2 protein levels (low in dry seeds) transiently increase within the first 12h-24h of imbibition and then decrease (see Fig. 1D condition "normal" of Piskurewicz et al 2008, same reference as above). When GA levels are low (as in PAC-treated seeds or as in *ga1* mutants), RGL2 protein is stabilized, which increases RGL2 levels as early as 12h upon imbibition, and this leads to 1) persistent RGL2 accumulation over time (see Fig. 1D condition "PAC" of Piskurewicz et al 2008).

This is indeed the case in both the endosperm and embryo (Fig. 3B and 3C). Since we are not sure whether this addresses your concern, we repeated the experiment in Fig. 3b and measured RGL2 protein levels in WT and *phyB* embryos and endosperms at earlier time points upon imbibition (6h, 12h, 24h) at 22°C and 28°C (Figure F). At those times, there are no signs of testa or endosperm rupture. These data are provided as western blot repetitions in Supplementary Figure 6.

Figure F

Protein gel blot analysis of RGL2 levels in WT and *phyB* embryos (upper panel) and endosperms (lower panel) from seeds cultivated under white light (50 $\mu\text{mol}/\text{m}^2/\text{s}$) at 22 and 28°C and dissected at the indicated times upon seed imbibition. Numbers represent RGL2 levels normalized to those of UGPase.

The most striking observation is that when seeds are thermoinhibited (i.e. *phyB* mutants at 28°C), endospermic RGL2 protein accumulation increases as early as 12h, which is consistent with the notion that they have lower GA levels. Over time, they will further increase RGL2 levels in both the endosperm and embryo (Figure 3b) consistent with the notion that thermoinhibited seeds have low GA levels.

Please note that depending on the time point and depending on the migration of the gel blot, one may detect one or two bands corresponding to RGL2 (see also supplemental Figure 15A of Piskurewicz et al 2008).

We have also repeated the experiment shown in Figure 3c and measured RGL2 protein in WT endosperms and embryos at earlier time points upon imbibition (12h, 24h, 48h) at 22°C and 34°C (Figure G). RGL2 protein accumulation increases as early as 12h in WT embryo, and most drastically in WT endosperm. As above, over time RGL2 will accumulate in thermoinhibited seeds in both the endosperm and embryo (Figure 3c). These data are provided as western blot repetitions in Supplementary Figure 6.

Figure G

Protein gel blot analysis of RGL2 levels in WT embryos (upper panel) and endosperms (lower panel) from seeds cultivated under white light (50 $\mu\text{mol}/\text{m}^2/\text{s}$) at 22 and 28°C and dissected at the indicated times upon seed imbibition. Numbers represent RGL2 levels normalized to those of UGPase.

4. Similarly, authors need to prove that the expression of PIF3 at lowT but not at highT is not associated with the developmental change.

We are sorry but we are not sure to understand why the expression of PIF3 should not be associated with the developmental change (see also our response to comment #4 by reviewer 1).

Under normal temperatures (22°C) that lead to germination, the embryonic protein accumulation of PIF3 upon imbibition does not exhibit marked changes at 6h, 12h and 24h (Figure H), then it strongly increases between 24h and 36h, remains high at 48h and then decreases (see Fig. 5a). At 24h the embryo has not germinated yet being still surrounded by the endosperm whereas at 36h the radicle as emerged, and the hypocotyl subsequently further elongates. Thus, the increase in PIF3 accumulation correlates with the developmental stage during which the hypocotyl elongates. Furthermore, the same pattern of embryonic PIF3 accumulation is observed in endosperm-less WT embryos at 22°C and 32°C (Fig. 5c). We found that in all these conditions *pif3* mutants end up having a shorter hypocotyl relative to WT (suppl Fig2d and Fig. 4b). These genetic results imply that PIF3 promotes hypocotyl elongation. The burst of PIF3 accumulation may reflect its biological function to promote elongation, i.e. the increase in PIF3 levels during the developmental phase where elongation takes place suggests that its absolute amounts matter for PIF3-mediated promotion of elongation. However, this is not an obligation as one could imagine that PIF3 could promote hypocotyl elongation without this involving an increase in its abundance during the elongation phase.

In thermoinhibited seeds, embryonic PIF3 accumulation remains low over time, and we showed that this is likely due to high ABA levels (Fig. 5a, Fig. 5c and Fig. 5d). Hence, in thermoinhibited seeds, embryonic PIF3 levels are kept low, which is consistent with the developmental state of embryonic growth arrest, a state where no elongation takes place (in contrast PIF3 levels increase in the endosperm under high temperatures, see our response to comment #4 by reviewer 1).

5. RNA-seq was performed at 48 hours after the imbibition. Authors wrote there is no observable difference between WT and *pif3* at this time point. However, if we magnify seeds in the supplemental figure 3, we can see that the *pif3* mutant seeds already undergo testa rupturing whereas WT seeds do not. Thus, the huge difference in RNA-seq data of the *pif3* mutant also could be due to the developmental stage difference.

We meant that there is no observable difference from the point of view of germination, which is radical protrusion out of the seed coat. Perhaps our wording was not precise enough [we wrote “We compared the transcriptome of WT and *pif3* endosperm from seeds imbibed for 48h at 30°C, i.e. at a time when WT and *pif3* seeds are developmentally indistinguishable as no *pif3* germination as yet taken place (Supplementary Figure 3a).”]. The main purpose of the RNAseq was to explore whether the genes that are deregulated in the *pif3* endosperm prior to germination are distinct from those deregulated in the embryo.

Our data suggest that this is indeed the case. That testa rupture is already visible in the *pif3* seeds at 48h is consistent with the notion that PIF3 plays a role in the endosperm under high temperatures. Indeed, testa rupture is likely the result of micropylar endosperm expansion (Figure 5 of De Giorgi et al 2015, PLOS Genetics Dec 17;11(12):e1005708). Thus, PIF3 appears to repress micropylar endosperm expansion under high temperatures.

We repeated the experiment shown in the supplementary Figure 3a of the first manuscript version at 24h, i.e. a time where no testa rupture is visible in both WT and *pif3* seeds (new supplementary fig 3a). As for 48h, we found that PIF3 regulates largely different sets of genes in the endosperm and embryo. These results further support the conclusion stated in the manuscript (“Altogether, these data support the conclusion that at high temperatures endospermic PIF3 has a distinct function, i.e. to promote thermoinhibition, relative to that of embryonic PIF3, i.e. to promote hypocotyl elongation in germinating seeds”).

We have changed the text of the results section to incorporate these new data (which are found in Supplementary Figure 3a together with the data for 48h).

6. It is difficult to follow author’s logic of claiming that PIF3 is the most important PIF mediating highT for seed germination. Germination phenotype of *pif1*, *pif3* and *pif5* single mutant is very similar (Fig 4a, Supple Fig 2). Authors also showed similar germination phenotypes of *pif1/pif3*, *pif3/pif4* and *pif1/pif3/pif5* multiple mutants. To claim that PIF3 is the major PIF, author need to show that *pif1/5* and *pif1/4* mutants have lower germination frequencies at highT than *pif3/5* and *pif3/4* mutants.

It was not our intention to convey that PIF3 is the most important PIF (e.g. we wrote “Altogether, these data show that PIFs, and notably PIF3, promote seed thermoinhibition. Together with the observation that *pifq7* mutants are thermoinhibited at very high temperatures (new supplementary Figure 2a), our results indicate that DELLA factors and PIFs promote endospermic ABA accumulation and release through parallel pathways.”. Reviewer 1 (point 2) had a similar remark. We take into consideration your remark and have modified the text of the title, the abstract, results and legends to ensure that we do not convey the wrong message. We paste here our reply to Reviewer 1:

To avoid conveying that PIF3 is the only PIF promoting thermoinhibition, we have modified the text of the title, the abstract, some parts of the results and the legend of the model to avoid potential misunderstandings (e.g. we now write “Altogether, these data show that PIFs, and notably PIF1, PIF3 and PIF5, promote seed thermoinhibition.”). The reason we think PIF3 is interesting and deserves a more detailed characterization is because it was 1) not previously involved in temperature responses, 2) not previously involved in the control of seed germination (unlike PIF1) and 3) it promotes antagonistic early embryonic growth responses in the endosperm and the embryo, which is not the case for other PIFs, particularly at high temperatures (Figure 4b). These arguments are now explicitly stated in the results section to justify our focus on studying PIF3 function during thermoinhibition.

7. [Minor] *pifp* mutant is written as *pifq* mutant.

Thank you, we corrected this and we name this mutant *pifq7* (see also our reply to reviewer 2 (Minor point 3)).

8. [minor] please show comparing immunoblot in the same blot (e.x. Fig 5a-endosperm, PIF3 at 22 and 34, Fig 5d) or quantify.

All the protein accumulation data shown in main figures and supplementary figures arise from the same blot. See also our reply to reviewer 4 (point 2): we have quantified the signals in the immunoblots and provided repetitions in supplementary data.

Reviewer #4 (Remarks to the Author):

In this manuscript, Piskurewicz and coworkers studied the role of the endosperm in thermoinhibition, or inhibition of seed germination at elevated temperature. They monitored germination of embryos with or without the endosperm (seed coat bedding assay), and used various mutants affected in phytochrome and hormone signaling. They demonstrated that thermoinhibition is mediated by inactivation of PhyB in the endosperm, which promotes accumulation of endospermic PIF3. PIF3 increases endospermic ABA synthesis, which leads to inhibition of embryo growth. In the embryo, elevated ABA inhibit PIF3, which would otherwise promote embryo growth.

The authors have used seed coat bedding assays in the past to dissect the role of the endosperm in seed germination. In this study, they have uncovered a prominent role for the endosperm in promoting thermoinhibition through phytochrome and ABA/GA signaling. The study is well done, and the results are interesting, however there is a large overlap with previous findings.

The positive and negative roles of ABA and GA synthesis/signaling in thermoinhibition, as well as the role of far-red light in inhibition of germination by elevated temperature, have been previously shown. Furthermore, the role of endospermic ABA in seed germination inhibition has been previously shown, although at control temperature. Previous work also showed that the endosperm inhibits germination under FR light and the same authors used a seed coat bedding assays to dissect the role of Phy and ABA signaling in the embryo and endosperm, although at control temperature. Phytochromes were also shown to play a role in seed germination across various temperatures. Last, the role of elevated temperature in promoting Phy inactivation was shown during hypocotyl growth. Overall, these findings decrease the novelty of this study.

REFS (also included in this papers): Hershchel et al 2007; Toh et al., 2008; Piskurewicz et al 2009; Lee et al 2010, 2012; etc.

We think that the main novelty of this work is to propose an integrated molecular genetic model describing satisfactorily how seed thermoinhibition is implemented in Arabidopsis: from the perception of high temperature by phyB to the downstream signaling components that promote endospermic ABA accumulation while describing the essential role of the endosperm in this context. To our best knowledge, an integrated model for seed thermoinhibition backed by experimental evidence was not previously described. Given the range of processes underlying thermoinhibition, this work would indeed have not been possible have it not been from the previous work that you refer to and that we also refer to in the manuscript. That body of work may make the thermoinhibition model described here foreseeable; however, there are other bodies of work in the literature that could lead us to foresee alternative models for temperature perception and signalization in the context of thermoinhibition (see for example comment by reviewer 2 who wonders whether ELF3

plays a thermosensing role). Our work provides experimental evidence for at least the model described here.

As you noted, other aspects of this work are worth mentioning such as defining endospermic PIF3 function and contrasting it with that of embryonic PIF3 (a similar argument could be made with PIF1). This result further explains the apparent paradox concerning PIFs whereby PIFs promote growth in seedlings whereas some PIFs repress embryonic growth in seeds. Furthermore, that the SCBA can be used in the context of seed thermoinhibition is methodologically significant. Lastly, we hope that this work will facilitate investigating or interpreting thermoinhibition in other species as thermoinhibition is relevant in ecology, phenology and agriculture.

Other comments:

1) Although RNAseq shows a minimal overlap in genes regulated by PIF3 in embryo and endosperm, suggesting different roles of PIF3 in these tissues, no follow-up of the comparative transcriptomic analysis is done aside from showing gene ontology enrichment. Therefore, it is unclear what these results mean without molecular, genetic, and functional validation.

The RNAseq is indeed just meant to further illustrate that endospermic PIF3 plays a distinct role from that of embryonic PIF3 (see also the new RNAseq data we provided in response to Reviewer 3, point 5). As mentioned above, the work is mainly focused on proposing a basic framework of how thermoinhibition works. The model describes the role of endospermic DELLAs and PIFs to promote endospermic ABA accumulation and release. Clearly, much needs to be done to understand how DELLAs and PIFs promote ABA endospermic ABA accumulation, this remains largely an open field of investigation that we feel is beyond the scope of this work. The RNAseq did, however, pick up *CYP707A1* as a potential PIF3 target to promote ABA accumulation (endospermic *CYP707A1* expression is upregulated in *pif3* mutants suggesting that PIF3 promotes endospermic ABA accumulation by repressing *CYP707A1* expression). We found that 2-month-old WT, *pif3*, *cyp707a1* and *pif3 cyp707a1* fully germinated 3 days upon imbibition at 22°C (Figure I). In contrast, *cyp707a1* were fully thermoinhibited at 28°C, unlike WT seeds, strongly suggesting that *CYP707A1* plays a significant role to regulate thermoinhibition, likely by limiting ABA levels in seeds. Furthermore, *pif3 cyp707a1* seeds were also fully thermoinhibited at 28°C (Figure I). We also observed that *cyp707a1* and *pif3 cyp707a1* thermoinhibition at 28°C required the presence of the endosperm (Figure I). Given that *pif3* seeds are less thermoinhibited at 30°C relative to WT (Figure 4a), these results show that *cyp707a1* is epistatic to *pif3*, suggesting that *CYP707A1* functions downstream of PIF3 in the endosperm as proposed by the model. These results are presented in Supplementary Figure 3c.

Figure I

2) Changes (or lack of) in protein levels need to be normalized and quantified in replicated experiments. Ideally, when comparing protein levels in different mutant backgrounds or under different growth conditions (temperature), protein extracts should be loaded on the same blot. All blots shown are cropped and it is difficult to compare protein levels. Examples: Fig 2d; 3b,c,f and 5a,c,d.

Although the western blots are cropped, all the protein extracts in a given panel were loaded on the same gels. We have provided repetitions in supplementary figure 6. Protein levels in the western blots are now quantified and normalized to those of UGPase. All the uncropped blots are provided in supplementary figure 7.

3) phyB/della and pif3/della mutants generated by CRISPR/Cas9: which generation (T2/T3) of mutants was characterized? was Cas9 was crossed out? Were PHYB transcripts analyzed in mutants?

The T3 generation was analyzed after crossing out Cas9 in the T2 generation (this information is added in the materials and methods section of the revised manuscript). We did not analyze *PHYB* transcripts but western blots show that phyB and PIF3 are undetectable in *della/phyB* and *della/pif3* mutants, respectively. Figure J shows PIF3 and phyB protein levels in 48h *della/phyB* and *della/pif3* seedlings grown in darkness (#1 and #2 are repetitions). These data are now included in the revised manuscript.

Reviewer #1 (Remarks to the Author):

I appreciate the revised manuscript is obviously improved and basically answer my comments, however, I still worry about the SCBA method, the hypocotyl length is the appropriate index to evaluate seed germination under high temperature, my comments are as below:

1) the author used jaz1 mutant to explain the JA signal is not functional during SCBA assay. Though JAZ1 is important regulator for JA signal, JAZs contain several components, thus I think just using jaz1 does not conclude that JA signal does not affect the seed germination during SCBA assay. At least using the triple or quadruple mutant of JAZs should be better than the single jaz1 mutant. In other words, I still worry whether SCBA is an efficiently assay to detect the real biological function of seed endosperm during seed germination.

2) using the hypocotyl length to detect the seed germination ability for embryo removing endosperm. As I suggested before, the phyb mutant shows long hypocotyl, which can not evaluate the seed germination. For example, Figure 2F, the phyB(G564E) showed high seed germination under HT, but the hypocotyl is still shorter, which means that the short hypocotyl for phyB(G564E) embryo without endosperm conflict the high seed germination. Therefore, other possible phenotype besides long hypocotyl should be used to evaluate the seed germination under high temperature

Reviewer #2 (Remarks to the Author):

The authors answered all my questions but I have a few comments on the revised version of their manuscript and the rebuttal letter.

1. On line 72f, the authors write "Inactive phyB promotes ABA synthesis ...". This sounds as if inactive phyB had a function. If this were the case, ABA synthesis should not be promoted in a phyB mutant, i.e. lack of phyB and phyB being in the Pr state would not have the same effect. However, I don't think that this is what the authors mean. They possibly mean that inactive phyB no longer prevents ABA synthesis. Another point: is it really ABA synthesis or ABA accumulation?

2. In the rebuttal letter, the authors speculate about "... a phytochrome that has evolved to no longer sense light but instead has evolved to be in equilibrium between an active and inactive signaling conformation with the ratio of active-to-inactive pool being set by temperature.". The temperature-sensing mechanism of phyB depends on the temperature-dependency of dark reversion, which requires activation of phyB by Pr-to-Pfr photoconversion. Therefore, temperature sensing by phyB is only possible in continuous light or light/dark cycles but not in continuous darkness. The mechanism proposed by the authors is different and a Pr/Pfr equilibrium would also exist in the dark. I do not know how likely this mechanism is to evolve. However, I am happy with the authors response. The comment on this point in my reviewer comments was out of curiosity, not concern.

3. The authors write in the discussion "This suggests that the OTHER phytochromes promote seed germination at 22°C under normal conditions and high temperatures would also accelerate their reversion to their inactive state in phyB mutants exposed to 28°C thus leading to phyB mutant thermoinhibition.". I was asking for a reference for this statement and the authors now suggest Jung et al., 2016, Science, 354, 886–889 and Legris et al., 2016, Science, 354, 897–900. These references show that dark reversion of phyB plays a role in temperature sensing. However, I was not clear enough when I asked for a reference and actually meant a reference showing a temperature effect on dark reversion of OTHER phytochromes. The supplemental information to Burgie et al., 2021, Proc. Natl. Acad. Sci. U. S. A., 118, e2105649118 contains such data (though not for all temperatures and in part only for phytochrome fragments – but maybe still a good reference in addition to Jung et al. and Legris et al.).

4. Supplementary Figure 6: It appears that Figure 1e shows one membrane for ABI5 while three different membranes are shown for UGPase. This means that data for ABI5 and UGPase are not

from the same lane from the same gel and therefore the quantification does not make sense. Please clarify this problem.

5. As I understand, Supplementary Figure 6 is meant to show replicates of blots shown in different figures. However, in some cases, conditions were not identical, e.g. time points are different for Figure 3c, the higher temperature is different for Figure 5c. However, I think that conditions should be identical for replicates.

6. Supplementary Figure 7: Uncropped membranes are missing for several figures, e.g. 1e, 2e, 2f, 3f (possibly 3d in Supplementary Figure 7 should be 3f), 5a (antibody control), and supplementary figures, e.g. 1c, 5.

7. Figure 3c/Supplementary Figure 7: On the uncropped membranes for Figure 3c (Embryo, End+), the signal for UGPase is much stronger at 22 °C than at 32 °C while it is similar at 22 °C and 32 °C in Figure 3c. This would mean that background/contrast was adjusted differently for the two parts derived from the same blot. Please double-check if this is the case. If it is the case, I think this should be mentioned. Please also confirm that this does not affect the quantification.

8. Figure 5d/Supplementary Figure 7: It appears that the part of the membrane showing PIF3 in phyB @ 22 °C in Figure 5d corresponds to the part of the membrane showing PIF3 in WT @ 28 °C in Supplementary Figure 7. Please double-check and correct if required.

Reviewer #3 (Remarks to the Author):

Authors clarified the most of my previous concerns. However, I still have a concern on the timing of transcriptome analysis and its interpretation.

1. On the SCBA assay. Although I agree with the utility of SCBA assay, I'm not quite sure if the experiment in Figure E addresses the specificity of endosperm. Authors are using 'ruptured' endosperm, whereas they are using seemingly intact petal, root, and cotyledon in Figure E. Since the effect is likely to be done by molecules secreted out to immersed solution, it is uncertain if such molecules are readily secreted out in intact organs. Chopped organs might mimic the ruptured endosperm. However, I agree with authors saying that, even if the effect is not specific to endosperm, only endosperm is right next to the embryo.

2. On transcriptome data. The testa rupture is a part of germination process and the radicle protrusion simply signifies the completion of seed germination, thus, I don't think authors can say there is 'no observable difference from the point of view of germination' or 'WT and pif3 seeds are developmentally indistinguishable as no pif3 germination as yet taken place'. In contrast to authors' description, I think WT and pif3 seeds are developmentally quite distinguishable. WT seeds that are germinating typically undergo testa rupture around at 24 hours after the imbibition at 22 °C, thus, it is likely that the pif3 mutant seeds at 48 hours at 30 °C are similar to WT seeds at 24 hours at 22 °C. Therefore transcriptomes presented in the manuscript are those of non-germinating WT seeds and pif3 mutant seeds in the middle of germinating rather than 'prior to germination'. Authors need to carry out RNA-seq much earlier to compare transcriptomes of developmentally (or at least morphologically) indistinguishable WT and pif3 seeds.

Reviewer #4 (Remarks to the Author):

I appreciate the revisions provided by the authors, however they have partially answered my previous comments/concerns:

1) RNAseq. I still believe that the RNAseq data could be further analyzed to support the model and/or uncover different mechanisms. For example, only DEGs related to ABA metabolism in the

endosperm at 48h were shown in supplement data 1. How did ABA and GA synthesis/signaling genes change in expression in the different seed compartments (embryo and endosperm) at different times? Also, which other regulators of germination/dormancy were differentially affected in embryo/endosperm? Ultimately, it is not clear from the current (minimal) analysis how the RNAseq data support the conclusion that "at high temperatures endospermic PIF3 has a distinct function, i.e. to promote thermoinhibition, relative to that of embryonic PIF3, i.e. to promote hypocotyl elongation in germinating seeds". Only numbers of DEGs were listed in the paper, and a very limited description was provided in the discussion about the biological processes or molecular functions enriched. A Figure on GO function enrichment should be presented and cluster analysis could be used to highlight patterns of DEGs. Verification by qPCR of selected DEGs is also usually shown.

2) Protein quantification. Typically, averages of three biological replicates are shown (with statistical analysis). This is especially important when showing changes in protein levels, as it is done for RNA levels in qPCR. It is not easy to compare all the different blots with quantifications as shown in the revised manuscript.

3) Details of CRISPR mutants were provided, thank you.

REVIEWER COMMENTS

We would like to thank the reviewers for their effort in evaluating our revision and for their constructive comments and criticisms. We have carefully read and addressed all the comments made by the reviewer in this second revision, which further improved the manuscript. In blue you will find the original comments made by the reviewers while our response is in black.

Reviewer #1 (Remarks to the Author):

I appreciate the revised manuscript is obviously improved and basically answer my comments, however, I still worry about the SCBA method, the hypocotyl length is the appropriate index to evaluate seed germination under high temperature, my comments are as below:

1) the author used *jaz1* mutant to explain the JA signal is not functional during SCBA assay. Though JAZ1 is important regulator for JA signal, JAZs contain several components, thus I think just using *jaz1* does not conclude that JA signal does not affect the seed germination during SCBA assay. At least using the triple or quadruple mutant of JAZs should be better than the single *jaz1* mutant. In other words, I still worry whether SCBA is an efficiently assay to detect the real biological function of seed endosperm during seed germination.

In the first revision we showed that under normal temperatures (22°C) a SCBA using WT endosperm and WT embryos does not block embryonic growth. So, under normal temperatures any JA signal triggered by the dissection procedure is not sufficient to block embryonic arrest. On the other hand, we observe WT embryonic growth arrest at high temperatures (34°C). The hypothesis you raise is that at high temperatures a JA signal could trigger embryonic growth arrest. In the first revision, we showed that WT and *jaz1* endosperm are similarly able to repress WT embryonic growth. In retrospect, we think this is not the best experiment to test the hypothesis. Indeed, mutating JAZ genes will lead to an overactive jasmonate signaling pathway in plants. Multiple *jaz* mutants have a constitutive defense response and plants are stunted and stressed. So, we do not think this is the appropriate genetic background to test the hypothesis because mutating multiple JAZ genes can potentially create an artificial situation of abnormally overactive JA signaling responses that potentially creates neomorphic effects in embryonic growth in the context of the SCBA that are unrelated to the hypothesis we wish to test.

To test your hypothesis, we think it is more appropriate to use mutants defective in JA biosynthesis or defective in JA signaling. If a JA signal is responsible for the growth arrest at high temperatures in the context of the SCBA, then at high temperatures the endosperm of mutants defective in JA biosynthesis or defective in JA signaling should have a weaker capacity to arrest embryonic growth relative to WT endosperm.

We therefore used *aos* mutants, which do not synthesize JA and therefore cannot initiate a JA signal. *aos* mutants are male sterile but can be propagated after spraying JA exogenously (Park JH et al. *Plant J* 31, 1-12 (2002)). Concerning JA signaling, a null *coi1* mutant (lacking COI1 through which jasmonoyl isoleucine (JA-Ile) signals) is problematic because mutants

are male sterile and cannot be rescued by exogenous JA. Hence the mutation must be propagated with Hz plants, and it is difficult to identify the mutant seeds in a segregating population. Therefore, we used a weak *coi1* allele (*coi1-34*, Acosta IF et al *Proc Natl Acad Sci U S A* 110, 15473-15478 (2013)), which is partially male sterile, i.e. it is able to produce seeds.

Hence, we used these mutants to test the hypothesis. WT, *coi1-34* and *aos* seeds fully germinated at 22°C after 3 days of imbibition (Figure below). At 34°C, WT, *coi1-34* and *aos* seeds were fully thermoinhibited, indicating that a defective JA synthesis or signaling does not affect seed thermoinhibition. Furthermore, WT, *coi1-34* and *aos* endosperms were similarly able to repress WT embryonic growth in a SCBA at 34°C indicating that endospermic JA synthesis and signaling does not play a major role in the context of the SCBA to promote WT embryonic arrest at high temperatures. We also checked the ability of WT endosperms to arrest *coi1-34* and *aos* embryos in a SCBA at 34°C and we found that growth of *coi1-34* and *aos* embryos was arrested similarly to that of WT embryos. Hence, embryonic JA synthesis and signaling does not play a major role in the context of the SCBA to promote embryonic arrest at high temperatures. Altogether, these experiments do not support the hypothesis that a JA signal promotes the embryonic growth arrest observed at high temperatures in the context of the SCBA.

a. Average germination percentage of WT, *aos* (*aos*) and *coi1-34* (*coi1*) seeds cultured 3 days under white light (50 $\mu\text{mol}/\text{m}^2/\text{s}$) at 22°C and 34°C (three replicates, $n=50$ seeds). b. Hypocotyl length was measured three days after imbibition at 34°C as indicated. "WT End-" "aos End-" and "coi1 End-" is the hypocotyl length of the WT, *aos* and *coi1-34* embryos, respectively, cultured without endosperm (dissection 4h after seed imbibition). For the SCBAs, the WT, *aos* (*aos*) and *coi1-34* (*coi1*) dissected embryos were cultured on a bed of WT or *aos* (*aos*) or *coi1-34* (*coi1*) endosperms as indicated. In a) and b) lower case letters above histograms are used to establish whether two values are statistically significantly different as assessed by one-ANOVA followed by a Tukey HSD test ($p<0.05$): different letters denote statistically different values.

2) using the hypocotyl length to detect the seed germination ability for embryo removing endosperm. As I suggested before, the *phyb* mutant shows long hypocotyl, which can not evaluate the seed germination. for example, Figure 2F, the *phyB(G564E)* showed high seed germination under HT, but the hypocotyl is still shorter, which means that the short hypocotyl for *phyB(G564E)* embryo without endosperm conflict the high seed germination. Therefore, other possible phenotype besides long hypocotyl should be used to evaluate the seed germination under high temperature

In the manuscript, we use hypocotyl length in two situations.

1) One is in the context of the SCBA where we monitor the activity of different endosperm genotypes to repress the growth of a WT embryo (e.g. Fig. 1b, Fig. 2c; Fig. 3a etc). When the WT hypocotyl is short it means the endosperm is blocking growth and when the hypocotyl is long it means that the endosperm is not blocking growth. Therefore, the WT hypocotyl length is used as a marker for endospermic activity to release ABA. In this context, the western blots monitoring ABI5 accumulation in the embryos are meant to complete the analysis.

2) Another situation (the one you referred to) is that concerned with seed material bearing mutations in phyB. In this case, we use hypocotyl length to emphasize the role that endospermic phyB has to promote thermoinhibition relative to the role of embryonic phyB to regulate embryonic growth, i.e. hypocotyl elongation.

Hence, in Fig 2f, we use phyB(G564E), which bears an amino acid change that slows down phyB thermal reversion, i.e. phyB(G564) remains active for a longer time relative to phyB. We expect therefore that the hypocotyl of phyB(G564E) embryos will be shorter than those containing phyB and we show that this is indeed the case. The point we want to make is that phyB(G564) seeds are able to germinate better under high temperatures than phyB seeds despite the fact that phyB(G564) embryos have a lower propensity to elongate relative to phyB embryos. This supports the notion that lower seed thermoinhibition in the phyB(G564) lines reflects weaker endosperm-imposed germination arrest.

Conversely, in Fig 2e the phyB(S86D) seeds germinate less at 30°C despite the fact that phyB(S86D) embryos grow faster than phyB embryos. This is consistent with the notion that the stronger thermoinhibition in the phyB(S86D) line reflects stronger endosperm-imposed germination arrest.

Reviewer #2 (Remarks to the Author):

The authors answered all my questions but I have a few comments on the revised version of their manuscript and the rebuttal letter.

1. On line 72f, the authors write "Inactive phyB promotes ABA synthesis ...". This sounds as if inactive phyB had a function. If this were the case, ABA synthesis should not be promoted in a phyB mutant, i.e. lack of phyB and phyB being in the Pr state would not have the same effect. However, I don't think that this is what the authors mean. They possibly mean that inactive phyB no longer prevents ABA synthesis. Another point: is it really ABA synthesis or ABA accumulation?

Thank you for picking this up (again). Indeed, you already raised this point in the first revision, and we agreed our phrasing lacked precision. We changed the manuscript accordingly (it is indeed ABA accumulation rather than synthesis), but we omitted to modify the phrasing in this part of the introduction. The phrasing is now corrected and consistent with other parts of the text. (We wrote "phyB inactivation leads to increased endospermic ABA levels and release, which blocks germination". We mean ABA accumulation not ABA synthesis.

2. In the rebuttal letter, the authors speculate about "... a phytochrome that has evolved to no longer sense light but instead has evolved to be in equilibrium between an active and inactive signaling conformation with the ratio of active-to-inactive pool being set by temperature.". The temperature-sensing mechanism of phyB depends on the temperature-dependency of dark reversion, which requires activation of phyB by Pr-to-Pfr photoconversion. Therefore, temperature sensing by phyB is only possible in continuous light or light/dark cycles but not in continuous darkness. The mechanism proposed by the authors is different and a Pr/Pfr equilibrium would also exist in the dark. I do not know how likely this mechanism is to evolve. However, I am happy with the authors response. The comment on this point in my reviewer comments was out of curiosity, not concern.

We also do not know how likely such a mechanism would evolve. Could instead the phytochrome itself have evolved from a protein sensing temperature (i.e. a protein in equilibrium between an active and inactive signaling conformation with the ratio of active-to-inactive pool being set by temperature) ?

3. The authors write in the discussion "This suggests that the OTHER phytochromes promote seed germination at 22°C under normal conditions and high temperatures would also accelerate their reversion to their inactive state in phyB mutants exposed to 28°C thus leading to phyB mutant thermoinhibition.". I was asking for a reference for this statement and the authors now suggest Jung et al., 2016, Science, 354, 886–889 and Legris et al., 2016, Science, 354, 897–900. These references show that dark reversion of phyB plays a role in temperature sensing. However, I was not clear enough when I asked for a reference and actually meant a reference showing a temperature effect on dark reversion of OTHER phytochromes. The supplemental information to Burgie et al., 2021, Proc. Natl. Acad. Sci. U. S. A., 118, e2105649118 contains such data (though not for all temperatures and in part only for phytochrome fragments – but maybe still a good reference in addition to Jung et al. and Legris et al.).

Sorry that we missed your point and thank you for your clarification and for providing this reference, which is now included in the text.

4. Supplementary Figure 6: It appears that Figure 1e shows one membrane for ABI5 while three different membranes are shown for UGPase. This means that data for ABI5 and UGPase are not from the same lane from the same gel and therefore the quantification does not make sense. Please clarify this problem.

Sorry for the confusion, the UGPase signals come from the same membrane and this membrane is the same as the one used for ABI5. We provide an uncropped image for the UGPase signals. Please note that Supplementary Figure 6 is now supplementary figure 8 in this second revision.

5. As I understand, Supplementary Figure 6 is meant to show replicates of blots shown in different figures. However, in some cases, conditions were not identical, e.g. time points are different for Figure 3c, the higher temperature is different for Figure 5c. However, I think that conditions should be identical for replicates.

The experiments in Supplementary Figure 6 (now supplementary figure 8 in this second revision) are meant to show biological replicates (not technical replicates) this is why in some cases the time points and temperatures are different. We think that showing different

time points or different temperatures (both 34°C and 32°C induce thermoinhibition in WT) lead to similar patterns in protein abundance strengths the validity of our observations and ultimately strengthen the model. Nonetheless, we specifically addressed your concerns. The cases where time points or temperatures differ are the following:

a) Figure 3b showed endospermic RGL2 levels for 24h at 48h (22°C and 28°C). In supplementary Fig 6 (now supplementary figure 8 in this second revision), endospermic RGL2 levels were shown for 6h, 12h and 24h.

=> We repeated the western blot at 24h and 48h.

b) Figure 3c showed endospermic and embryonic RGL2 levels at 48h and 72h (22°C and 32°C). In supplementary Fig 6 (now supplementary figure 8 in this second revision), endospermic and embryonic RGL2 levels were missing for 72h.

=> We repeated the western blots at 48h and 72h.

c) Figure 5a, showed embryonic PIF3 levels at 24h, 36h, 48h, 72h and 96h (22°C and 34°C). In supplementary Fig 6 (now supplementary figure 8 in this second revision), embryonic PIF3 levels were missing at 96h.

=> We repeated this western blot including the 96h time point.

Furthermore, Figure 5a showed endospermic PIF3 levels at 48h and 72h (22°C and 34°C). In supplementary Fig 6 (now supplementary figure 8 in this second revision), endospermic PIF3 levels were missing at 72h.

=> We repeated this western blot including the 72h time point.

d) Figure 5c showed a time course of embryonic PIF3 accumulation at 32°C whereas the supplementary Fig 6 (now supplementary figure 8 in this second revision) showed a time course at 34°C. We repeated the western blot at 32°C.

e) Figure 5d had no repetition.

=> we have repeated this experiment which is now in supplementary Fig 8. This figure is mainly meant to show that at 28°C, *phyB* mutant embryos from *phyB* thermoinhibited seeds have low PIF3 levels relative to *phyB* embryos deprived of their endosperm (see our comments to point 2 of reviewer 4).

6. Supplementary Figure 7: Uncropped membranes are missing for several figures, e.g. 1e, 2e, 2f, 3f (possibly 3d in Supplementary Figure 7 should be 3f), 5a (antibody control), and supplementary figures, e.g. 1c, 5.

These figures show signals arising from the same membrane. We just selected the region of interest from that membrane. But it seems that you want us to show a bigger region of the blot. This is now provided in supplementary figure 9 where we provide bigger regions of the blots for 1) the main figures, 2) the blot in supplementary Fig. 1c, 3) the blot in supplementary Fig. 5 and 4) the blots of the repetitions in supplementary Fig. 8.

7. Figure 3c/Supplementary Figure 7: On the uncropped membranes for Figure 3c (Embryo, End+), the signal for UGPase is much stronger at 22 °C than at 32 °C while it is similar at

22 °C and 32 °C in Figure 3c. This would mean that background/contrast was adjusted differently for the two parts derived from the same blot. Please double-check if this is the case. If it is the case, I think this should be mentioned. Please also confirm that this does not affect the quantification.

We did not adjust the background/contrast differently in different parts of the same blot so the quantification will not be affected. We agree with you that the two images do not quite fit in terms of signal intensities. We acquire several images showing different exposures. It looks that in supplementary Figure 7 (now supplementary figure 9 in this second revision) we did not use the exact same exposure as in Figure 3c. But there might be some other explanation related to image processing by the computer. To avoid this misrepresentation, in the new figure 3c we use the UGPase signal previously present in supplementary figure 7 (now supplementary figure 8 in this second revision). This experiment has also been repeated (see above).

8. Figure 5d/Supplementary Figure 7: It appears that the part of the membrane showing PIF3 in phyB @ 22 °C in Figure 5d corresponds to the part of the membrane showing PIF3 in WT @ 28 °C in Supplementary Figure 7. Please double-check and correct if required.

Thanks for pointing this out. The Supplementary figure 7 was mislabeled. This is now corrected in supplementary figure 9. A similar problem with Figure 2d was also corrected.

Reviewer #3 (Remarks to the Author):

Authors clarified the most of my previous concerns. However, I still have a concern on the timing of transcriptome analysis and its interpretation.

1. On the SCBA assay. Although I agree with the utility of SCBA assay, I'm not quite sure if the experiment in Figure E addresses the specificity of endosperm. Authors are using 'ruptured' endosperm, whereas they are using seemingly intact petal, root, and cotyledon in Figure E. Since the effect is likely to be done by molecules secreted out to immersed solution, it is uncertain if such molecules are readily secreted out in intact organs. Chopped organs might mimic the ruptured endosperm. However, I agree with authors saying that, even if the effect is not specific to endosperm, only endosperm is right next to the embryo. When the endosperm ruptures upon germination, it ruptures only in one place (at the level of the micropylar endosperm). In our experiments the endosperm is dissected out from seeds that have not germinated yet (4h after seed imbibition). When we dissect, we perform a local small incision and then we gently squeeze out the embryo through the hole generated by the incision by applying a gentle pressure with blunted forceps. Therefore, the endosperm we use was briefly stretched while the embryo is squeezed out and bears a small incision. The endosperm is therefore not chopped and used as such to assemble the bed of endosperms. Similarly, in the experiments with the petals, roots and cotyledons, we are using materials dissected after performing a small incision and manipulating the tissues with a forceps. So, we think the experiments we reported with petals, roots and cotyledons are as equivalent as they can be as those we perform in the context of the SCBA.

2. On transcriptome data. The testa rupture is a part of germination process and the radicle protrusion simply signifies the completion of seed germination, thus, I don't think authors

can say there is 'no observable difference from the point of view of germination' or 'WT and *pif3* seeds are developmentally indistinguishable as no *pif3* germination as yet taken place'. In contrast to authors' description, I think WT and *pif3* seeds are developmentally quite distinguishable. WT seeds that are germinating typically undergo testa rupture around at 24 hours after the imbibition at 22 °C, thus, it is likely that the *pif3* mutant seeds at 48 hours at 30 °C are similar to WT seeds at 24 hours at 22 °C. Therefore transcriptomes presented in the manuscript are those of non-germinating WT seeds and *pif3* mutant seeds in the middle of germinating rather than 'prior to germination'. Authors need to carry out RNA-seq much earlier to compare transcriptomes of developmentally (or at least morphologically) indistinguishable WT and *pif3* seeds.

You initially correctly pointed out that *pif3* mutants showed testa rupture events at 30°C at 48h. In the previous revision we performed an RNAseq at 24h (30°C) and in this case WT and *pif3* are developmentally indistinguishable, including absence of testa rupture. We are not sure whether you missed this additional data. You might, however, think that we should look even earlier. However, we think that 24h is a good time point because the WT and *pif3* seeds are developmentally indistinguishable, including the total absence of testa rupture. By looking even earlier than 24h we might be probing differences that could be no longer relevant for the control of seed germination. Together with the data at 48h, the transcriptome datasets support the model that PIF3 regulates largely different sets of genes in the endosperm and embryo.

Reviewer #4 (Remarks to the Author):

I appreciate the revisions provided by the authors, however they have partially answered my previous comments/concerns:

1) RNAseq. I still believe that the RNAseq data could be further analyzed to support the model and/or uncover different mechanisms. For example, only DEGs related to ABA metabolism in the endosperm at 48h were shown in supplement data 1. How did ABA and GA synthesis/signaling genes change in expression in the different seed compartments (embryo and endosperm) at different times? Also, which other regulators of germination/dormancy were differentially affected in embryo/endosperm? Ultimately, it is not clear from the current (minimal) analysis how the RNAseq data support the conclusion that "at high temperatures endospermic PIF3 has a distinct function, i.e. to promote thermoinhibition, relative to that of embryonic PIF3, i.e. to promote hypocotyl elongation in germinating seeds". Only numbers of DEGs were listed in the paper, and a very limited description was provided in the discussion about the biological processes or molecular functions enriched. A Figure on Go function enrichment should be presented and cluster analysis could be used to highlight patterns of DEGs. Verification by qPCR of selected DEGs is also usually shown.

Our formulation that you quote ("..it is not clear [...] how the RNAseq data support the conclusion that "at high temperatures endospermic PIF3 has a distinct function, i.e. to promote thermoinhibition, relative to that of embryonic PIF3, i.e. to promote hypocotyl elongation in germinating seeds". ") is misleading and we apologize for the confusion. The

transcriptome analysis is not really meant to support the conclusion that endospermic PIF3 has a distinct function to promote thermoinhibition relative to that of embryonic PIF3, which is to promote hypocotyl elongation. Evidence for this conclusion was provided before genetically (Fig 4)(furthermore, PIF3 accumulation data in Fig 5 show that PIF3 is differentially regulated by temperature in the endosperm and embryo). Rather, what the transcriptome data convey is that PIF3 regulates different sets of genes in the endosperm and embryo, consistent with the notion that endospermic PIF3 could play distinct roles from those of embryonic PIF3. So, we reformulated our statement by writing “Altogether, these data are consistent with the notion that endospermic PIF3 could play distinct roles relative to those of embryonic PIF3”.

Again, what the transcriptome data convey is that PIF3 regulates different sets of genes in the endosperm and embryo. In this second revision (Supplementary Fig. 4a, 4b , 4c and Supplementary Data 1), we have revamped the presentation of the transcriptome data by providing 1) a better description and presentation of the gene ontology (GO) enrichment analysis and 2) the expression changes in genes bound by PIF3 (potentially direct targets of PIF3 according to Zhang et al. 2013 (Zhang et al. 2013, PLoS Genet e1003244) and in genes involved in the metabolism and signaling of ABA and GA. The transcriptome data can be used by researchers to further study the functions of PIF3 in the endosperm and the embryo. In the first version of the manuscript, we used these data to pinpoint *CYP707A1* whose expression is upregulated in the *pif3* endosperm relative to WT endosperm under high temperatures. In the first revision, we provided genetic evidence that *CYP707A1* functions downstream of PIF3 in the endosperm as initially proposed by the model. In this second revision, we verified by qPCR that *CYP707A1* expression is upregulated in the *pif3* endosperm under high temperatures, consistent with the transcriptome data.

2) Protein quantification. Typically, averages of three biological replicates are shown (with statistical analysis). This is especially important when showing changes in protein levels, as it is done for RNA levels in qPCR. It is not easy to compare all the different blots with quantifications as shown in the revised manuscript.

Our protein data are not meant to provide an exact quantification of protein abundance. Rather, they are meant to provide semi-quantitative data regarding the relative abundance of endogenous ABI5, RGL2, phyB and PIF3 protein levels under specific conditions.

ABI5 and RGL2 are protein markers of high ABA and low GA levels, respectively and the high accumulation of ABI5 and RGL2 in the context of thermoinhibition is clear and in line with other publications describing situations of high ABA and low GA levels, respectively, where seed germination is arrested (e.g. Piskurewicz et al 2008 Plant Cell Oct;20(10):2729-45; Piskurewicz et al 2008, EMBO J Aug 5;28(15):2259-71).

Concerning phyB, we provide its abundance between transgenic lines or between different genetic backgrounds to indicate that absolute phyB levels do not account for the observed phenotypes. For example, in Fig.2e we show that phyBS86D and phyB levels are similar in the phyBS86D line and the phyB line, respectively. Similarly, in Fig. 2f phyBG564E and phyB levels are similar in the phyBG564E line and the phyB line, respectively. In supplementary Figure 1c we wanted to show that phyB levels are not higher in *della* mutants under high

temperatures to exclude that the low thermoinhibition of *della* mutants is not due to high phyB accumulation. The data shows two time points and if anything, they indicate that phyB levels are lower in *della* mutants under high temperature. So, these data strongly suggest that the low thermoinhibition of *della* mutants cannot be the result of higher phyB accumulation (we wrote “At high temperatures, phyB accumulation was moderately lower in *della* mutant seeds relative to WT seeds indicating that enhanced phyB signaling that would result from high phyB levels in *della* seeds does not account for their lack of thermoinhibition (Supplementary Figure 1c)”

Concerning PIF3, there are three main messages we want to convey:

1) At 22°C, i.e. when seed germination is not blocked, or at high temperatures in embryos deprived of their endosperm, PIF3 accumulation increases markedly and transiently when the embryo is germinating (i.e. at 36h and 48h upon imbibition approximately). This can be clearly appreciated in Fig. 5a and Fig. 5c (+ biological replicates for Fig. 5a and Fig 5c shown in Supplementary Fig. 8). Fig. 5d (+ biological replicates for Fig. 5d shown in Supplementary Fig. 8) shows that at 28°C, *phyB* mutant embryos from *phyB* thermoinhibited seeds have low PIF3 levels relative to *phyB* embryos deprived of their endosperm.

2) At high temperatures that block seed germination or in embryos treated with ABA, the increase in PIF3 accumulation is no longer observed. This can be clearly appreciated in Fig. 5a and Fig. 5c (+ biological replicates for Fig. 5a and Fig 5c shown in Supplementary Fig. 8).

3) At high temperatures that block seed germination, PIF3 accumulation in the endosperm increases relative to low temperatures, i.e. an effect that is opposite to that observed in germinating embryos. This can be clearly appreciated in Fig. 5a (+ biological replicates for Fig. 5a shown in Supplementary Fig. 8).

3) Details of crisper mutants were provided, thank you.

Thank you again for picking this up.

Reviewer #1 (Remarks to the Author):

Thank the authors for their continuous efforts to explain my questions in this version, I have not more questions about it, though I STILL the role of PIF1 should be efficiently evaluate during seed thermoinhibition, like PIF3.

Reviewer #2 (Remarks to the Author):

I thank the authors for their detailed response and the additional data they provided. In doing so, they have addressed all my questions and concerns. I like the manuscript very much and hope it can be published soon!